# Genetic linkage disequilibrium of deleterious mutations in threatened mammals

Chunyan Hu [ID][1,2,5], Gaoming Liu [ID][1,5], Zhan Zhang[1,2], Qi Pan[1,2], Xiaoxiao Zhang [ID][1,2], Weiqiang Liu [ID][1,2], Zihao Li[1,2], Meng Li [ID][1], Pingfen Zhu [ID][1], Ting Ji[1], Paul A Garber [ID][3,4] & Xuming Zhou [ID][1✉]

## Abstract

The impact of negative selection against deleterious mutations in endangered species remains underexplored. Recent studies have measured mutation load by comparing the accumulation of deleterious mutations, however, this method is most effective when comparing within and between populations of phylogenetically closely related species. Here, we introduced new statistics, LDcor, and its standardized form nLDcor, which allows us to detect and compare global linkage disequilibrium of deleterious mutations across species using unphased genotypes. These statistics measure averaged pairwise standardized covariance and standardize mutation differences based on the standard deviation of alleles to reflect selection intensity. We then examined selection strength in the genomes of seven mammals. Tigers exhibited an over-dispersion of deleterious mutations, while gorillas, giant pandas, and golden snub-nosed monkeys displayed negative linkage disequilibrium. Furthermore, the distribution of deleterious mutations in threatened mammals did not reveal consistent trends. Our results indicate that these newly developed statistics could help us understand the genetic burden of threatened species.

**Keywords** Negative Selection; Deleterious Mutations; Linkage Disequilibrium; Threatened Mammals
**Subject Category** Evolution & Ecology

## Introduction

According to the IUCN red list, ~27% of assessed species are threatened with extinction, with mammals accounting for 25% of all endangered species (IUCN 2022). Assessing genetic diversity of these species is a critical indicator of a species' evolutionary potential, and can help to identify the set of conservation priorities needed to protect threatened taxa (Schläpfer and Schmid, 1999; Loreau et al, 2001; Hoban et al, 2021). In this regard, the development of high-throughput sequencing technologies has

enabled conservation biologists to examine genetic diversity on a genome-wide scale. In the case of several threatened species such as the chimpanzee (*Pan troglodytes*), Tasmanian devil (*Sarcophilus harrisii*), tiger (*Panthera tigris*), lion (*Panthera leo*), snow leopard (*Panthera uncia*), two species of snub-nosed monkeys (*Rhinopithecus spp.*), finless porpoises (*Neophocaena spp.*) and giant panda (*Ailuropoda melanoleuca*), their heterozygosity or genetic diversity is not significantly lower than that of many non-threatened species including humans (Cho et al, 2013; Zhang et al, 2015; Liu et al, 2018a; Zhao et al, 2013; Zhou et al, 2016). This has led researchers to question the sensitivity of current estimators of genetic diversity in gauging genetic burden and to reconsider the presumed correlations between population size and genetic diversity.

Because many threatened species are characterized by extended periods of low population size, they are likely to experience less efficient negative selection and greater genetic drift, resulting in the accumulation of more deleterious mutations. Recent studies tend to assess the mutation burden by identifying deleterious mutations, which decrease fitness of carriers in general, and thus may be prevented by negative selection especially in genomes of threatened species (Dussex et al, 2021; Xue et al, 2015; Xie et al, 2022). For example, by counting the frequency of deleterious variants, Xue et al, 2015 revealed that mountain gorillas (*Gorilla gorilla beringei*, estimated remaining population of 2600 mature individuals, IUCN 2023) have relatively fewer deleterious variants than western lowland gorillas (*Gorilla beringei graueri*, estimated remaining population of >3800 mature individuals, IUCN 2023) (Xue et al, 2015). Similarly, Xie et al, 2022 have counted and compared the average number of derived deleterious and missense variants in the genomes of the threatened crocodile lizard (*Shinisaurus crocodilurus*). They found that the strongly deleterious alleles were purged effectively in this inbred lizard species (Xie et al, 2022).

These methodologies have enabled researchers to identify selection intensity and genetic burden in animal populations. However, the effects of deleterious mutations in comparative analyses should be homologous and have similar consequences, especially among different populations of the same species as well as in phylogenetically closely related species, which limit its implementation in cross-species comparisons. The magnitude of linkage disequilibrium (LD), i.e., nonrandom associations among loci, between deleterious mutations has attracted attentions recently because it is directly related to the mutation load paradox, and a

[1]Key Laboratory of Animal Ecology and Conservation Biology, Institute of Zoology, Chinese Academy of Sciences, 100101 Beijing, China. [2]University of Chinese Academy of Sciences, 100049 Beijing, China. [3]Department of Anthropology, Program in Ecology, Evolution, and Conservation Biology, University of Illinois, Urbana, IL, USA. [4]International Center of Boidiversity and Primate Conservation, Dali University, Dali, China. [5]These authors contributed equally: Chunyan Hu, Gaoming Liu. ✉E-mail: zhouxuming@ioz.ac.cn

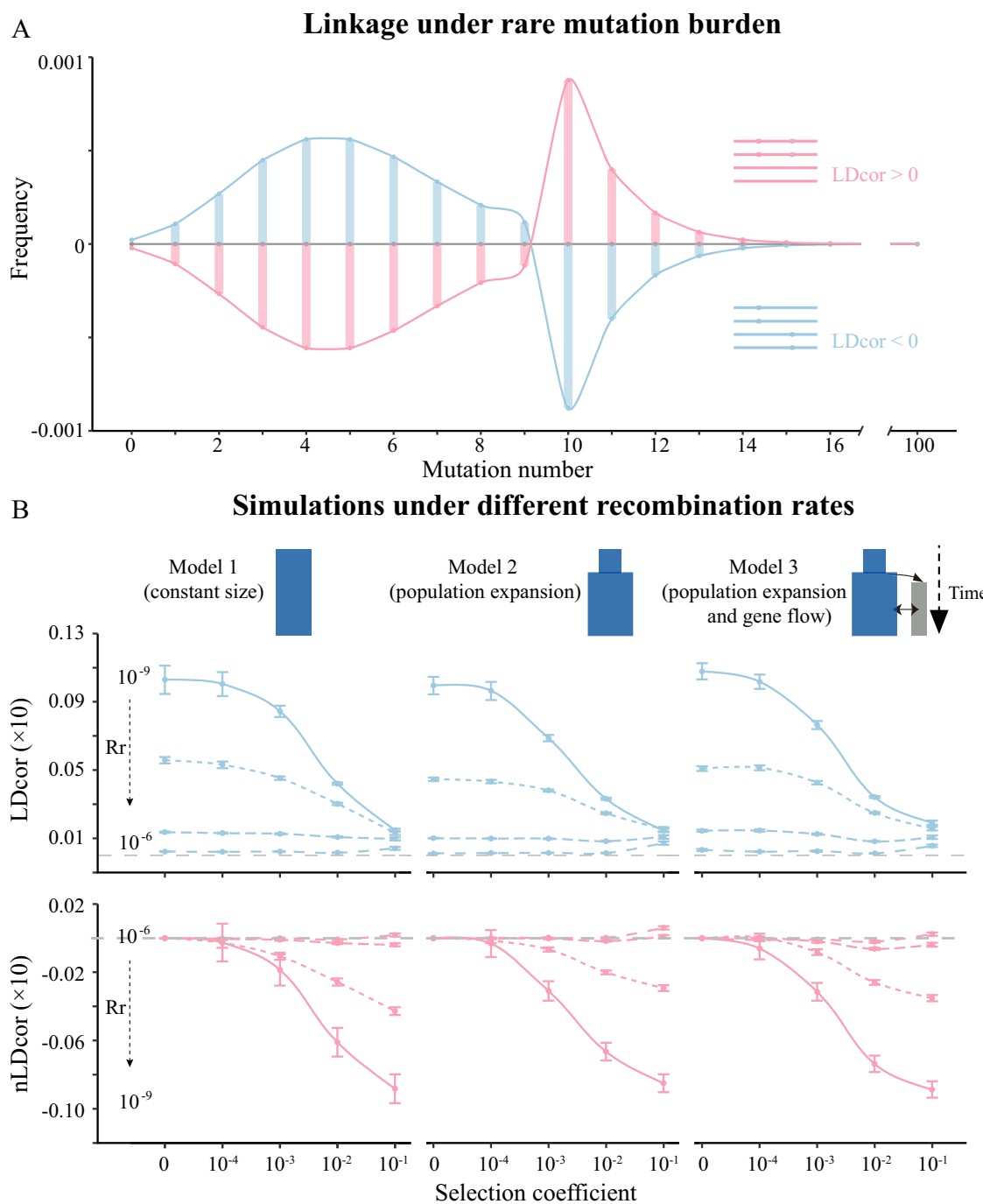

**Figure 1. LDcor and nLDcor with different selection coefficients, recombination rates, and demographic models.**

(A) The mutation burden (normalized by subtracting the frequency of multiplicative selection (LDcor = 0)) of variants under antagonistic epistasis (LDcor > 0), and synergistic epistasis (LDcor < 0). (B) We tested three different models (model 1: constant population size; model 2: population expansion; model 3: population expansion and gene flow). We did 120 simulations for each recombination rate (Rr, values in $10^{-9}$, $10^{-8}$, $10^{-7}$, or $10^{-6}$) and negative selection coefficient (values in 0, $10^{-4}$, $10^{-3}$, $10^{-2}$, or $10^{-1}$). The points and the error bars represent the mean value and the s.e.m. of LDcor and nLDcor.

species' fitness landscape and demographic history (Muller, 1950; Charlesworth, 1990, 1998; Kondrashov, 1998; West et al, 1998). Negative selection may shape linkage disequilibrium patterns of deleterious mutations with synergistic epistasis and Hill-Robertson Interference (HRI). Under synergistic epistasis, individuals carrying multiple

deleterious variants experience a greater decrease in relative fitness than the summation of each variant, and are less competitive in surviving or breeding. Negative selection with synergistic epistasis removes haplotypes containing multiple deleterious alleles, leading to negative LD (Garcia and Lohmueller, 2021) (Fig. 1A). Synergistic epistasis can

alleviate the mutation burden (Crow and Kimura, 1979; Kimura and Maruyama, 1966; Kondrashov, 1988), and make sexual reproduction favorable (Kondrashov, 1988). In this regard, Sohail et al, 2017 found that rare deleterious mutations are located further away from each other than expected by chance and that the detected under-dispersion of rare loss-of-function alleles was present in the genomes of humans and flies (Sohail et al, 2017). Recently, a consistent accelerated inbreeding depression for fitness was found in two different fly populations (Domínguez-García et al, 2019), suggesting synergistic epistasis among deleterious alleles. Finally, a second way in which negative selection can affect LD is through HRI, which has demonstrated that linkage among loci will dampen the effectiveness of selection and is more likely to influence those pairs of variants close to each other (Hill and Robertson, 2007; Garcia and Lohmueller, 2021). Therefore, the LD of deleterious mutations can be used to quantify selection intensity and, more importantly, the degree to which its magnitude is comparable across species.

Here, we introduce two new statistics of LD magnitude: the averaged pairwise standardized covariance (LDcor) and its standardized form (nLDcor) for all mutations, and explore relationships between the global LD magnitude and selection intensity. LDcor is consistent with classical $r^2$ and $D$' (signed by $D$) but genotype phasing is not required. We then examine the LD magnitude of deleterious mutations in the genomes of several endangered or threatened mammals, as well as in humans and one domestic mammal (sheep). These analyses and results suggest that assessing the LD magnitude of deleterious mutations is a promising indicator of negative selection. This approach can help establish conservation priorities and design effective strategies to protect a taxonomically diverse array of threatened mammals when used alongside other conservation data and observations.

# Results

## Negative selection is correlated with LD between pairs of doubletons

The new statistics (LDcor and nLDcor) proposed in this research are computed using a diploid genotype, and are compared among different populations to quantify whether derived alleles are more likely to be coupled (i.e., present in the same individuals) or in repulsion (i.e., present in different individuals). LDcor for weakly and moderately deleterious variants ($s = -0.0001$, $s = -0.001$, $s = -0.01$) are lower than those for neutral doubletons. Given that nLDcor is usually negative, this indicates that deleterious doubletons ($s < 0$) tend to occur in different individuals (Fig. 1). This result holds even if the allele frequency spectrum is controlled (Appendix Figs. S1 and S2). In addition, under conditions when other parameters are unchanged, LDcor and nLDcor decrease monotonically as the negative selection coefficient increases. Similarly, LDcor was found to generally increase monotonically as the recombination rate decreases. The increment of LDcor decreases monotonically as the negative selection coefficient rises. Finally, the absolute value of nLDcor was found to generally increase monotonically as the recombination rate decreases. However, LD can be ignored under a strong selection coefficient or a high recombination rate, which is consistent with the scenario proposed by Kimura (Kimura, 1965) and Nagylaki (Nagylaki, 1974, 1976).

We also tested recessive variants, and found that pairs of deleterious SNPs ($s = -0.001$, $-0.01$, or $-0.1$) tend to have lower mean values of LDcor and negative values of nLDcor under conditions of high recombination rates ($r \geq 1 \times 10^{-7}$), as well as higher mean values of LDcor compared to values for neutral SNPs as well as positive values of nLDcor under conditions of low recombination rates ($r \leq 1 \times 10^{-8}$) (Appendix Fig. S3).

## Effects of epistasis, selection, bottlenecks, and demographics on LD

To examine how the LDcor and nLDcor of variants are impacted by epistasis level, we simulated different synergistic epistasis levels, as well as different negative selective coefficient for each model using a recombination rate of $1 \times 10^{-8}$ crossovers per base pair per generation (Fig. 2A; Appendix Fig. S4A). For any selection coefficient, LDcor consistently decreases monotonically as the level of synergistic epistasis increases. For nLDcor, there is an opposite effect. Under the same synergistic epistasis level, both LDcor and nLDcor typically decrease monotonically as the negative selection coefficient increases.

We next tested the effect of varying magnitudes of bottlenecks and resampling based on model 1 (Fig. 2B; Appendix Fig. S4B). We tried different extents of population rescaling as well as different duration times. For any specified rescaling (take a subset of 10,000, 1000, 700, or 400) and duration (keep 0, 2, 6, or 10 generations for bottleneck), LDcor and nLDcor decrease with an increasing negative selection coefficient. In addition, LDcor increased and nLDcor decreased as rescaled size decreased and as the duration lengthened.

We next examined the potential effects of different demographic models (model 1: constant population size; model 2: population expansion; model 3: population expansion and gene flow) and species population sizes (Fig. 2C; Appendix Fig. S4C). In simulations based on 10,000 individuals, LDcor and nLDcor decreased monotonically as the negative selection coefficient increased in all three models. For majority specific selection coefficients ($-10^{-3} \sim 0$), model 1 had the largest LDcor value and the smallest nLDcor value, followed by model 3 and model 2. This may be due to the population expansion experienced by the latter two models. Model 3 shows the largest LDcor value and smallest nLDcor value when population size was set to 1000 or 100 for most tested selection coefficient. This may be caused by strong genetic drift.

The maximum derived allele count (e.g., MDAC), which limits demographic information, also was found to affect LDcor (Appendix Fig. S5). Un conditions in which a large number of individuals are evolving (e.g., 10,000 or 1000 individuals) and taken (1000 individuals), LDcor correlated with the selection coefficient when the MDAF was within a measurable range (e.g., MDAF < = 5%). Nevertheless, with 100 individuals evolving in the population and the same number sampled, the correlation between LDcor and the selection coefficient resembled the pattern seen in simulations in which more variants were taken (e.g., MDAF > 5%). This may be caused by uncertainties along with drift.

Since negative selection with synergistic epistasis is more likely to impact pairs of variants in the genome (Sohail et al, 2017), whereas other factors such as the Hill-Robertson effects act on pairs of variants that are close to each other (Garcia and Lohmueller,

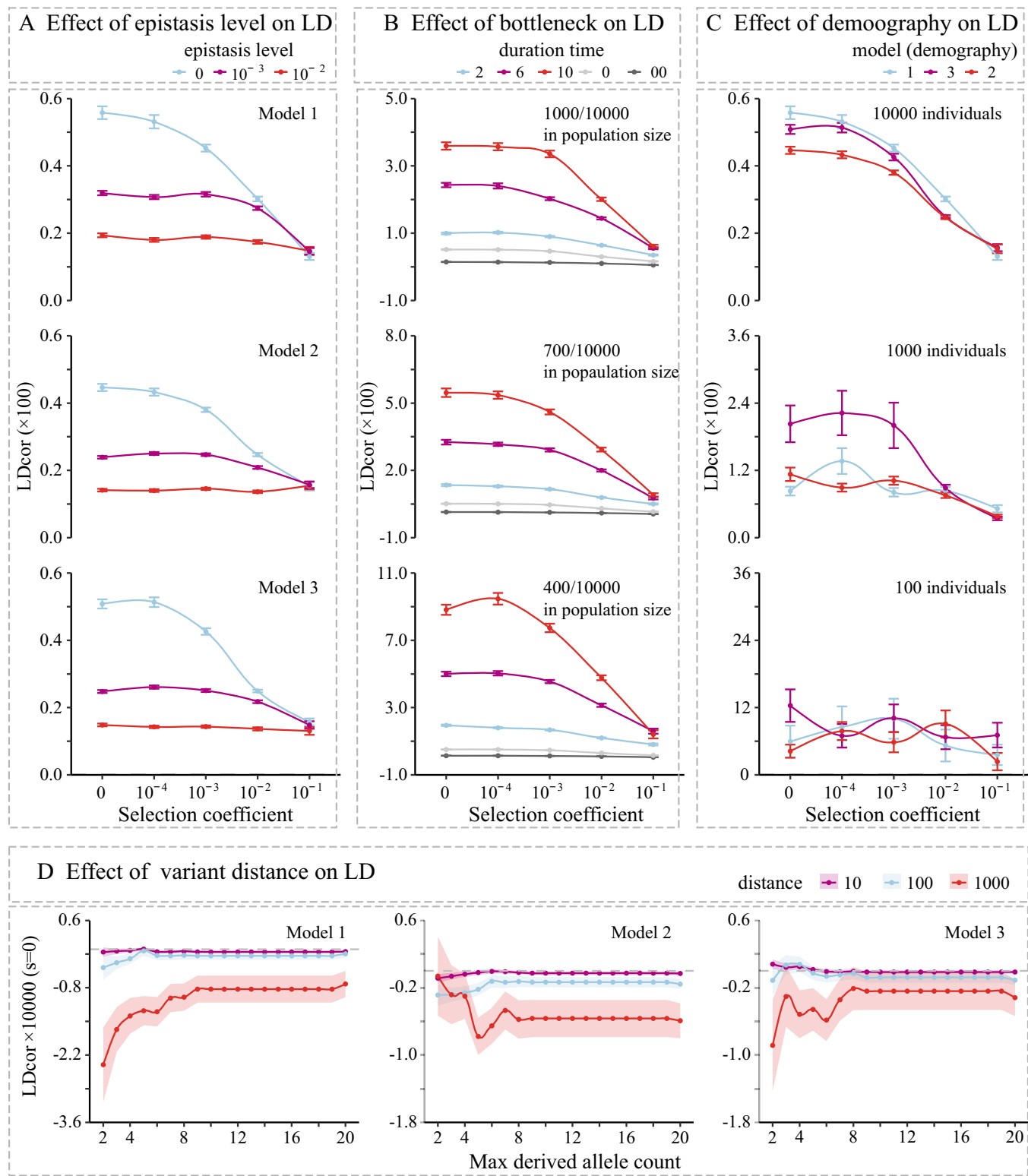

A Effect of epistasis level on LD

B Effect of bottleneck on LD

C Effect of demoography on LD

D Effect of variant distance on LD

2021), we also explore the main source of negative LD detected by LDcor among variants irrespective of whether they were synonymous or non-synonymous variants. Simulations under these three demographic models showed that LDcor is usually negatively correlated with loci distance, although the correlation was not

significant (Fig. 2D; Appendix Fig. S6 and Appendix Tables S1 and S2). We also tested variants at certain distances from each other in several different human populations (Appendix Fig. S7). LDcor of variants that do not have very close distance (100 bp, 1000 bp) are usually smaller than that of all variants (i.e.,

◄ **Figure 2. LDcor with different levels of synergistic epistasis, bottlenecks, demography, and maximum derived allele frequency.**

(A) We simulated different synergistic epistasis levels (values in 0, $10^{-3}$, $10^{-2}$) as well as different negative selective coefficients (0, $10^{-1}$, $10^{-2}$, $10^{-3}$, and $10^{-4}$) for each model (10,000 individuals) with a recombination rate of $1 \times 10^{-8}$ crossovers per base pair per generation. The points and the error bars represent the mean value and the s.e.m. of LDcor. (B) In order to compare the LDcor of varying magnitudes of bottlenecks (based on model 1), we simulated different scaled population sizes (from 10,000 to 1000: bottleneck 1, 700: bottleneck 2, and 400: bottleneck 3) as well as different duration times (0, 2, 6, and 10 generations). The dark gray line is the results of no-resampling (10,000) at time 0 (e.g., time 00). The points and the error bars represent the mean value and the s.e.m. of LDcor. (C) The effect of different models and population sizes (sample 1 (10,000): a simulation with 10,000 individuals and a subset of 1000 individuals in calculating; sample 2 (1000): a simulation with 1000 individuals and a subset of 1000 individuals in calculating; sample 3 (100): a simulation with 100 individuals and a subset of 100 individuals in calculating on LDcor for different selection coefficients. (D) For all three models, we examined the LDcor of variants with selection coefficients of 0 and beyond a certain distance away from each other (10 bp, 100 bp, and 1000 bp). The LDcor values were normalized by reducing the LDcor of variants without distance limitations. The points and the error bars represent the mean value and the s.e.m. of LDcor.

0 bp). Meanwhile, LDcor is enhanced when loci located far apart (e.g., 10,000 bp) were examined (Appendix Figs. S6 and S7).

## The new LD statistics exhibit stability across various allele frequency spectrums

In order to compare the utility of our LD statistics, we computed classical LD statistics: $D$, $D'$, and $r^2$ and compared the four mean values of classical, signed $D'$, and $\sqrt{r^2}$ under different parameters. A high Pearson correlation ($R > 0.85$ with $P < 2.2 \times 10^{-16}$) was found for all comparisons between LDcor and signed $\sqrt{r^2}$ (e.g., for simulations with $1 \times 10^{-8}$) (Fig. 3A).

We then compared our new statistics with NetLD (Sohail et al, 2017), which also uses unphased data as input under varied MDAFs (Fig. 3B). There was no significant difference between LDcor and NetLD (Sohail et al, 2017) when the derived allele count were limited to 2. We also compared the three LD classical statistics under different MDAF values (by scaling LD mean and standard error mean based on that of dac = 2 separately). The results indicate that as MDAF expands, LDcor shows a decreasing mean standard error, whereas NetLD typically shows an increasing mean standard error. In addition, while LDcor was always found to decrease as MDAF increased, NetLD showed no consistent pattern. For example, NetLD increased for neutral or weakly deleterious variants ($s = 0$, $s = -0.0001$, $s = -0.001$) and decreased for moderately deleterious variants ($s = -0.01$). Overall, the overlapping distribution in NetLD when MDAF doubled (e.g., from 0.01 to 0.02), was considerably smaller than LDcor (LDcor = 0.96; NetLD = 0.40). These results highlight that LDcor is relatively stable, even when the allele frequency spectrum of the variant set was not controlled.

## Estimation of LD in seven mammalian species

To test whether LD patterns are affected by negative selection among mammals, we compared the whole-genome sequences of humans (or CEU), gorillas, giant pandas (or panda), sheep, golden snub-nosed monkeys (or goldenSM), tiger, and rhesus macaques. For each population, we computed the LD statistics on rare and global alleles for synonymous, missense, and loss-of-function (LoF) mutations (here defined as splice site disruptions or nonsense) (Table 1; Fig. 4). In addition, we tested the results while controlling for the derived allele frequency spectrum (Appendix Fig. S8). Our results showed that the LDcor values for synonymous and missense loci sets consistently exhibited over-dispersion (>0), while the LoF loci sets occasionally displayed under-dispersion. Upon further

examination of the LD metrics within these LoF loci sets in these mammals, we found that most populations had negative LDcor values, with the exceptions being the tiger, golden snub-nosed monkey, and sheep at the maximum MDAC level (which includes all biallelic variants with a derived allele count greater than 1). When considering global alleles, the nLDcor values for the LoF loci sets remained negative in all tested populations, except for the tiger, at the maximum MDAC level.

Some variations in LD metrics were observed when focusing on rare alleles. Within the sheep population, LDcor values exhibited significant over-dispersion in LoF and missense loci sets when compared to synonymous datasets. Conversely, for most non-domesticated species, we found that rare LoF allele sets (e.g., MDAC $\leq$ 10) showed under-dispersion relative to synonymous and missense sets sharing the same MDAC metrics. In the CEU populations, both LDcor and nLDcor values were negative and consistently the lowest within LoF variants at appropriate MDAC levels. This indicates that LoF experienced the least linkage load and were most strongly influenced by negative selection. This effect is evident not only in the selection coefficient but also in the combined impact of synergistic epistasis and other selection-related forces across variant categories. This trend was similarly observed in other species: gorillas (Critically Endangered), rhesus macaques (Least Concern), golden snub-nosed monkeys (Endangered), and giant pandas (Vulnerable). In gorillas, LDcor scores were generally the lowest and nLDcor were generally negative for LoF variants, with the exception at MDAC = 3. For rhesus macaques, LDcor and nLDcor values were positive and typically highest in LoF variants. In golden snub-nosed monkeys, both statistics were notably negative and lowest in LoF at MDACs of 3, 4, 5, or 6, but not at other ranges. Giant panda populations exhibit negative LDcor in LoF at an MDAC of 2 or 3. Similarly, we found a negative nLDcor at MDACs of 2, 3, 7, or 8, For giant pandas, there was a positive LDcor in LoF for MDAC > 4. In tiger populations, LDcor was always positive and nLDcor was generally negative in LoF (MDAC > 3).

We conducted simulations of the variants under the demographic histories of the giant panda and the golden snub-nosed monkey (Zhao et al, 2013; Zhou et al, 2016), as there was non-negligible changing when meeting a specific MDAC (Fig. 5). The simulations demonstrated that in the goldenSM, LDcor values decreased and nLDcor became negative for variants under higher negative selective pressure Alternatively, for variants with low selection coefficients ($s = -0.0001$), LDcor values aligned with those observed under neutral conditions ($s = 0$), and nLDcor values were not consistently negative. This suggests a potential linkage excess of

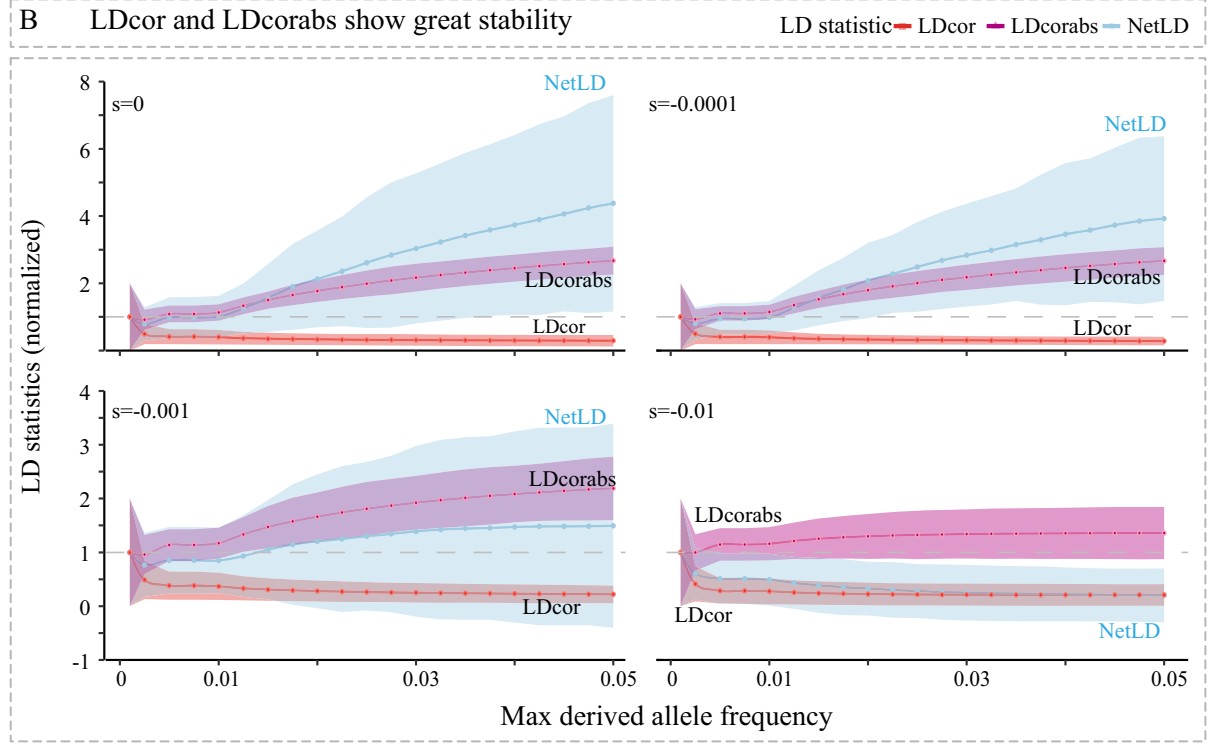

**Figure 3.  LDcor and LDcorabs show consistency with classical LD statistics.**

(A) For all three models (Model 1: constant population size; Model 2: population expansion; model 3: population expansion and gene flow) simulated with 10,000 individuals, we took a subset of 1000 individuals and tested the consistency of LDcor with the mean of the square root of classical $r^2$ (signed by D) (signed $\sqrt{r^2}$) for a loci set with a recombination rate of $1 \times 10^{-8}$ crossovers per base pair per generation under selection coefficients of 0. Only loci with a derived allele count no less than 5, 10, or 15, were included (maximum derived allele count, or MDAC). (B) The mean and the s.e.m. (normalized by the one with a derived allele count of 2, i.e., scaled to have mean = 1 and s.e.m. = 1 when mdac = 2) of three LD statistics varied as the maximum derived allele frequency increased under different selection coefficients (0, $10^{-4}$, $10^{-3}$, and $10^{-2}$).

mildly selected variants. In giant pandas, our results consistently showed that LDcor values were lower and nLDcor values were negative for variants under higher negative selective pressure, indicating a greater extent of negative linkage disequilibrium in LoF variants compared to synonymous variants. Neutral rare variants and weakly deleterious rare variants exhibited similar LD patterns in giant pandas. Furthermore, we conducted simulations run for 100 generations with the giant panda's population size scaled-down tenfold. The results indicated that the LDcor values of slightly deleterious variants were indistinguishable from those of neutral variants, and nLDcor values were not consistently negative (Fig. 5), suggesting a relaxation of negative selection.

## Discussion

### Advantages of LDcor and nLDcor

Genetic diversity is an important indicator used in conservation genetics to measure the evolutionary potential and fitness of a threatened population or species. However, measuring genetic diversity alone, ignores the importance of linkage among different loci, particularly the deleterious mutations that provide critical information concerning the genetic health of threatened species. The classical LD-detecting statistic $r^2$ and $D'$ depend on haplotype information, which is often difficult to obtain. This challenge has steered recent research towards analyses based on unphased data. This body of work generally falls into two approaches: one involves estimating haplotype information to then calculate traditional or modified traditional statistics (e.g., sigma$^2_d$ (Ragsdale and Gravel, 2020)), whereas the other bypasses the need for haplotypes entirely (for example, NetLD (Sohail et al, 2017) as well as $H_R^{(j)}$ (Garcia and Lohmueller, 2021)). However, these methods come with limitations. For example, estimations, particularly those relying on EM iterations, may not always pinpoint the global maximum, leading to potential inaccuracies (Gaunt et al, 2007).

In this study, we measured the correlation between diploid allele counts (as denoted by {0,1, or 2}) as an alternative approach to estimate LD. LDcor eliminates the need for haplotype information and incorporates variance differences among loci pairs, so as to prevent the effects of low-frequency mutations from being obscured by high-frequency mutations. Assuming that low-frequency mutations account either for a great majority or a small fraction of the totality of mutations, LDcor is more sensitive to low-frequency signals (Appendix Table S3).

Furthermore, when considering different populations within or between species that present a different allele frequency spectrum, we note that when using LDcor, non-neutral variants can be normalized based on data from neutral variants, which we call

nLDcor. After the neutral normalization, the false-positive correlations originating from limited and different sample sizes or other factors, can be partly offset. In short, nLDcor takes the LD among different loci into account, standardizes the differences among mutations with different allele frequencies, and normalizes the LD of deleterious loci with synonymous loci. Thus, nLDcor values are comparable among different populations and species.

### The effect of confounders on LD

It appears that LDcor is always negatively correlated with negative selection intensity and recombination rate under the additive model ($h = 0.5$; Fig. 1B). Negative selection induces negative LD and an insufficient recombination rate induces LD. Under the recessive model ($h = 0$; Appendix Fig. S3), this pattern becomes disordered. The observed LD pattern for recessive deleterious mutations may result from the complex interplay of heterosis effect (associative overdominance), where these mutations are masked from selection in their heterozygous state (Pamilo and Pálsson, 1998). This effect can be more pronounced if there is a low degree of recombination and the deleterious alleles occur on the same haplotype, leading to positive LDcor.

Synergistic epistasis under negative selection can enforce the power in bringing in negative LD (Figs. 2A and 4A). Both negative selection and synergistic epistasis can increase negative LD, suggesting that an individual locus may not be the appropriate unit of selection (Franklin and Lewontin, 1970; Slatkin, 1972). Genetic drift, which can magnify with founder effect and population bottlenecks, reduces genetic variation and increases LD (Rogers, 2014; Slatkin, 1994; Pritchard and Przeworski, 2001) (Fig. 2B,C; Appendix Fig. S4B,C). The largest LD values in model 1 (in simulation with 10,000 individuals and subset of 1,000 individuals) support the importance of founder effect on LD (Fig. 2C; Appendix Fig. S4C). The observed higher LD values in model 3 compared to model 2 are likely attributable to gene flow and variations in allele frequency across subpopulations, which are known factors that contribute to LD (Li and Nei, 1974; Slatkin, 1975; Mitton and Koehn, 1973; Nei and Li, 1973) (Fig. 2B; Appendix Fig. S4B). The observation that LD increases with both intensified population size scaling and appropriate duration immediately following population scaling underscores the impact of a population bottleneck on LD (Fig. 2C; Appendix Fig. S4C). The role of Hill-Robertson effects, which mainly describes the interface between sites separated by small distances, in negative LD would be limited because the removal of closely linked loci (within 100 bp or 1000 bp) resulted in smaller LDcor values (Fig. 2D).

Given that forward simulations included mutations affecting fitness multiplicatively across loci, we further surmise that high levels of negative LD between LoF SNPs can be attributed to

Table 1. Global linkage disequilibrium among different allele sets of importance in the tiger, rhesus macaque, golden snub-nosed monkey, giant panda, human (CEU), and sheep genomes.

| Species | Type | Nloci | Mean | LDcor | nLDcor | LDcorabs | NetLD |
|---------|------|-------|------|-------|--------|----------|-------|
| Tiger | LoF | 187 | 126.03 | 3.78E-03 | 3.75E-03 | 1.93E-01 | 2.32E-04 |
| | Missense | 10214 | 7789.91 | 5.86E-04 | 5.58E-04 | 1.94E-01 | 5.55E-05 |
| | Synonymous | 14941 | 13189.34 | 2.75E-05 | / | 2.02E-01 | 5.71E-07 |
| Macaque | LoF | 138 | 70.53 | −1.10E-03 | −1.21E-03 | 1.54E-01 | −5.48E-05 |
| | Missense | 5599 | 3208.75 | 3.47E-04 | 2.41E-04 | 1.53E-01 | 5.58E-05 |
| | Synonymous | 6055 | 3736.44 | 1.06E-04 | / | 1.53E-01 | 2.01E-05 |
| GoldenSM | LoF | 450 | 291.88 | 5.34E-04 | −3.91E-04 | 1.87E-01 | −6.69E-05 |
| | Missense | 17018 | 11346.56 | 3.14E-04 | −6.11E-04 | 2.01E-01 | 2.28E-04 |
| | Synonymous | 15068 | 9183.18 | 9.25E-04 | / | 2.25E-01 | 8.21E-04 |
| Panda | LoF | 59 | 40.69 | −5.45E-03 | −5.44E-03 | 1.29E-01 | −1.09E-03 |
| | Missense | 3421 | 2626.55 | 7.20E-05 | 8.56E-05 | 1.28E-01 | 2.27E-05 |
| | Synonymous | 3150 | 2737.94 | −1.36E-05 | / | 1.30E-01 | 8.62E-06 |
| CEU | LoF | 464 | 642.76 | −3.03E-04 | −3.14E-04 | 5.74E-02 | 9.09E-06 |
| | Missense | 21500 | 26990.12 | 1.63E-05 | 5.29E-06 | 6.74E-02 | 1.51E-06 |
| | Synonymous | 17964 | 20921.45 | 1.10E-05 | / | 7.36E-02 | 3.60E-06 |
| Sheep | LoF | 1882 | 2070.68 | 7.28E-05 | −3.36E-05 | 9.78E-02 | −4.45E-06 |
| | Missense | 41056 | 44015.57 | 3.38E-05 | −7.26E-05 | 9.94E-02 | −5.62E-07 |
| | Synonymous | 46596 | 49325.72 | 1.06E-04 | / | 1.01E-01 | 1.02E-06 |
| Gorilla | LoF | 275 | 323.55 | −2.48E-04 | −4.09E-04 | 1.47E-01 | −1.94E-04 |
| | Missense | 12005 | 13990.71 | 6.50E-05 | −9.56E-05 | 1.50E-01 | 7.77E-06 |
| | Synonymous | 12066 | 13652.1 | 1.61E-04 | / | 1.57E-01 | 1.89E-05 |

negative selection coupled with synergistic epistasis. However, a limitation of our study is that we did not consider the possibility that some mutations may not be unconditionally deleterious. Instead, these mutations could be subject to stabilizing selection rather than directional selection, which may also result in negative LD values (Charlesworth, 2013).

## Limitations of LDcor

Because LDcor decreases with the intensify of the selection coefficient, the expectation of LDcor should be almost zero under the null assumption of neutral sites without epistasis. It cannot be ignored that LDcor values of neutral sites are significantly greater than 0 in our simulations and experimental data. This can be caused by several factors including insufficient loci, batch effects, inadequate recombination rates, hitchhiking effects, and a cline in the average rare mutation burden. Even with negative LD and negative LDcor, there are several additional factors beyond selection coefficients and synergistic epistasis that can contribute to these patterns. Stabilizing selection on certain mutations can also result in negative LD. Furthermore, genetic drift, especially in small populations, can act as a random force creating LD. In addition, low-quality sequencing or improper filtering techniques may result in an insufficient number of loci and a significantly biased derived allele frequency spectrum, which can erroneously lead to conclusions of negative LD.

Furthermore, while the standardization of LDcor (i.e., nLDcor) was implemented to mitigate the impacts of hitchhiking, genetic drift, demographic factors, or data quality, the normalization process inherently presumes synonymity in mutations—assuming them to be neutral and uniform across instances. This assumption has increasingly come under scrutiny. As such, the development of future LD estimators will need to differentiate or measure the variability inherent in synonymous mutations.

## Empirical data interpretation

Although analyses of empirical data revealed that not all LDcor values scale inversely with the strength of selection pressure and not all nLDcor values are negative, the results provide insights into the conservation of threatened mammals. For instance, the gorilla population, characterized by a low rate of reproduction, exhibited the smallest LD statistics in LoF variants when compared to synonymous and missense variants under both maximum and small MDAC levels. This is similar to findings in humans and sheep. In the giant panda population, both LDcor and nLDcor of LoF variants were negative when the MDAC was maximum or no more than 4. These findings suggest that, despite some species having small population sizes and being listed as Endangered by the IUCN, their genomes currently exhibit a low linkage load of coinherited LoF mutations. In contrast, the tiger population exhibited positive LDcor and nLDcor values for LoF loci sets at both maximum and minimum MDAC levels (i.e., MDAC = 2). Therefore, greater attention should be given to the increased linkage burden in such species.

Nonetheless, it is essential to note that under the combined effects of selection and epistasis, populations are likely to survive

## LD statistics of some mammal populations

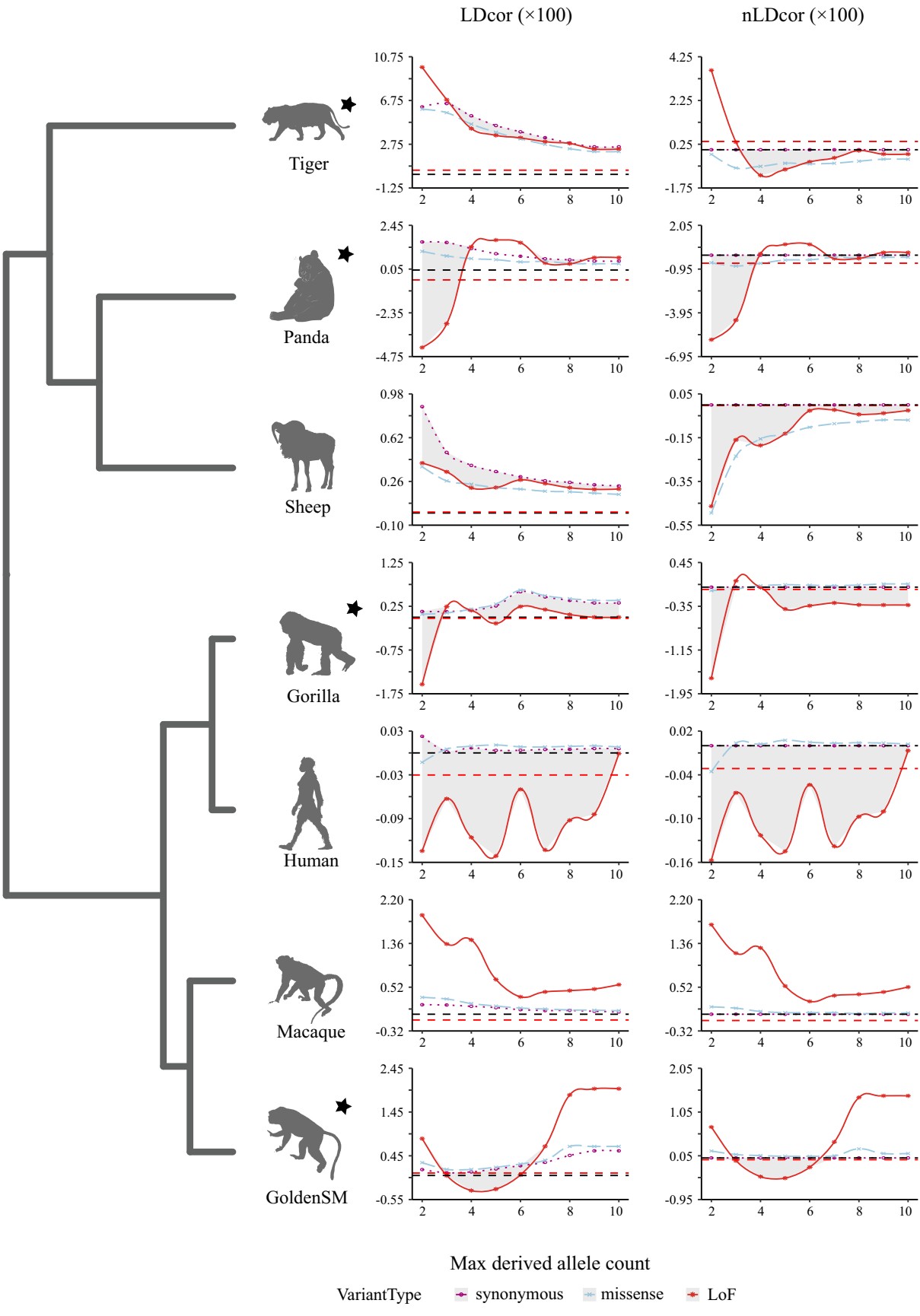

LDcor (×100)  nLDcor (×100)

Max derived allele count

VariantType ● synonymous ✳ missense ✳ LoF

**Figure 4. LD statistics for a select set of mammal populations.**

We classified alleles as synonymous, missense, or LoF (red) using Ensemble Variant Effect Predictor version 0.1.13 (McLaren et al, 2016). LD (LDcor, nLDcor, LDcorabs) for the three loci sets that were computed and compared for each mammal population. Gray shadows represent the lower LD in LoF relative to the synonymous set. Gray dotted lines represent "0". Red dotted lines represent the LD values of LoF loci sets when MDAC was maximum. The exact formulae of the two statistics are detailed in "Methods". Threatened species were identified with five-pointed stars.

for a long time even under high mutation rates (Sohail et al, 2017). Therefore, populations experiencing negative LDcor and nLDcor values may not be at a high genetic risk of extinction. However, positive LDcor and nLDcor values suggest the potential for a sharp decrease in population size and/or relaxation of selection pressures. These new indicators (LDcor and nLDcor), combined with our empirical observations and analyses focusing on individual fitness, have important theoretical and practical implications for species conservation.

## Methods

### Computing LD summary statistics

Three different pairwise LD statistics were used for comparison: $r^2$, $D'$, and NetLD (Sohail et al, 2017). The definition of $r^2$ and $D'$ as well as several other statistics are based on the classical D, which use haplotype frequency as a primary input. For a pair of diallelic loci i and j with allele A/a and B/b,

$$D_{AB} = p_{AB} - p_A * p_B$$

$$D_{AB} = -D_{Ab} = -D_{aB} = D_{ab}$$

$$r^2 = \frac{D_{AB}^2}{p_A * p_a * p_B * p_b}$$

$$D' = \begin{cases} \frac{D_{AB}}{max(-p_A*p_B, -p_a*p_b)} & D_{AB} < 0 \\ \frac{D_{AB}}{min(p_A*p_b, p_a*p_B)} & D_{AB} > 0 \end{cases}$$

Where $p_{AB}$ is the frequency of haplotype AB, $p_A$, $p_B$, $p_a$, and $p_b$ are the frequencies of allele A, B, a, and b.

Another LD statistic NetLD (Sohail et al, 2017) uses allele frequency as a primary input.

$$NetLD = \frac{2}{n(n-1)} \sum_{\substack{i<j \\ i,j \in L}} cov(Xi, Xj)$$

Where Xi and Xj are discrete random variables that represent the number of derived alleles present at loci i and j in a n-individual population and L is a locus set. Xi and Xj represent the derived allele count of loci i and j, and range in value from 0, 1, or 2. Covariance of Xi and Xj is denoted as cov (Xi, Xj). Then, V/V$_A$ was calculated as (n(n-1)/2*NetLD +V$_A$)/V$_A$, where V$_A$ represents the additive

variance of all variants in the tested set and V represents variance for the sum of all variants (Sohail et al, 2017).

The calculation of $r^2$ and $D'$ was done using PLINK 2.0 (Chen et al, 2019), and the calculation of NetLD (Sohail et al, 2017) was done using Rscript.

### New LD statistics for derived deleterious alleles

To measure the normalized LD of deleterious mutations, the averaged pairwise standardized covariance (LDcor) for all mutations (including missense, synonymous, LoF, etc.) were calculated as:

$$LDcor_L = \frac{2}{n(n-1)} \sum_{\substack{i<j \\ i,j \in L}} \frac{cov(Xi, Xj)}{\sigma(Xi) * \sigma(Xj)}, \quad (1)$$

$$LDcorabs_L = \frac{2}{n(n-1)} \sum_{\substack{i<j \\ i,j \in L}} \frac{|cov(Xi, Xj)|}{\sigma(Xi) * \sigma(Xj)}. \quad (2)$$

Where Xi and Xj are defined as above. The covariance of Xi and Xj are denoted as cov (Xi, Xj), and the standard deviation of Xi and Xj are denoted as σ(Xi) and σ(Xj). For each pair of loci (i, j), $\frac{cov(Xi,Xj)}{\sigma(Xi)*\sigma(Xj)}$ represents the linear correlation coefficient and ranges from −1 to 1. LDcor values range from −1 to 1, and LDcorabs values range from 0 to 1. LDcor equals to 0 when negative linkage disequilibrium exactly offsets positive linkage disequilibrium, and LDcorabs equals 0 if and only if all loci are independent. In these calculations, the inequality $LDcor_L <= LDcorabs_L$ always holds.

In the absence of selection and synergistic epistasis, variants affect the mutation burden independently and genetic variance should be additive (Kimura and Maruyama, 1966; Sohail et al, 2017). Both LDcor and LDcorabs among neutral variants are expected to be zero. When negative selection and synergistic epistasis are present, negative linkage disequilibrium appears, and LDcor gets closer to −1 and LDcorabs gets closer to 1. The exact value is greatly affected by the strength of selection and the extent of epistasis. Antagonistic (diminishing returns) epistasis, in contrast, creates positive LD between deleterious mutations and increases the LDcor (Fig. 1A).

The variance of loci i and j are computed as $p_A * p_a$ and $p_B * p_b$ resulting in $\frac{n(n-1)}{2} LDcorabs_L = \sum_{\substack{i<j \\ i,j \in L}} \left| \frac{cov(Xi,Xj)}{\sigma(Xi)*\sigma(Xj)} \right| = \sum_{\substack{i<j \\ i,j \in L}} \left| \frac{E(Xi*Xj)-E(Xi)E(Xj)}{\sigma(Xi)*\sigma(Xj)} \right|$, which in form is very similar to $2\sum_{\substack{i<j \\ i,j \in L}} \sqrt{\frac{D_{AB}^2}{p_A*p_a*p_B*p_b}} = 2\sum_{\substack{i<j \\ i,j \in L}} \sqrt{r_{ij}^2}$. $D'$ only differs from $\sqrt{r^2}$ (the sqrt of classical $r^2$) by taking different a standardization and a

## Simulations of infered history (A,B) and assumed history (C) under different selection coefficient

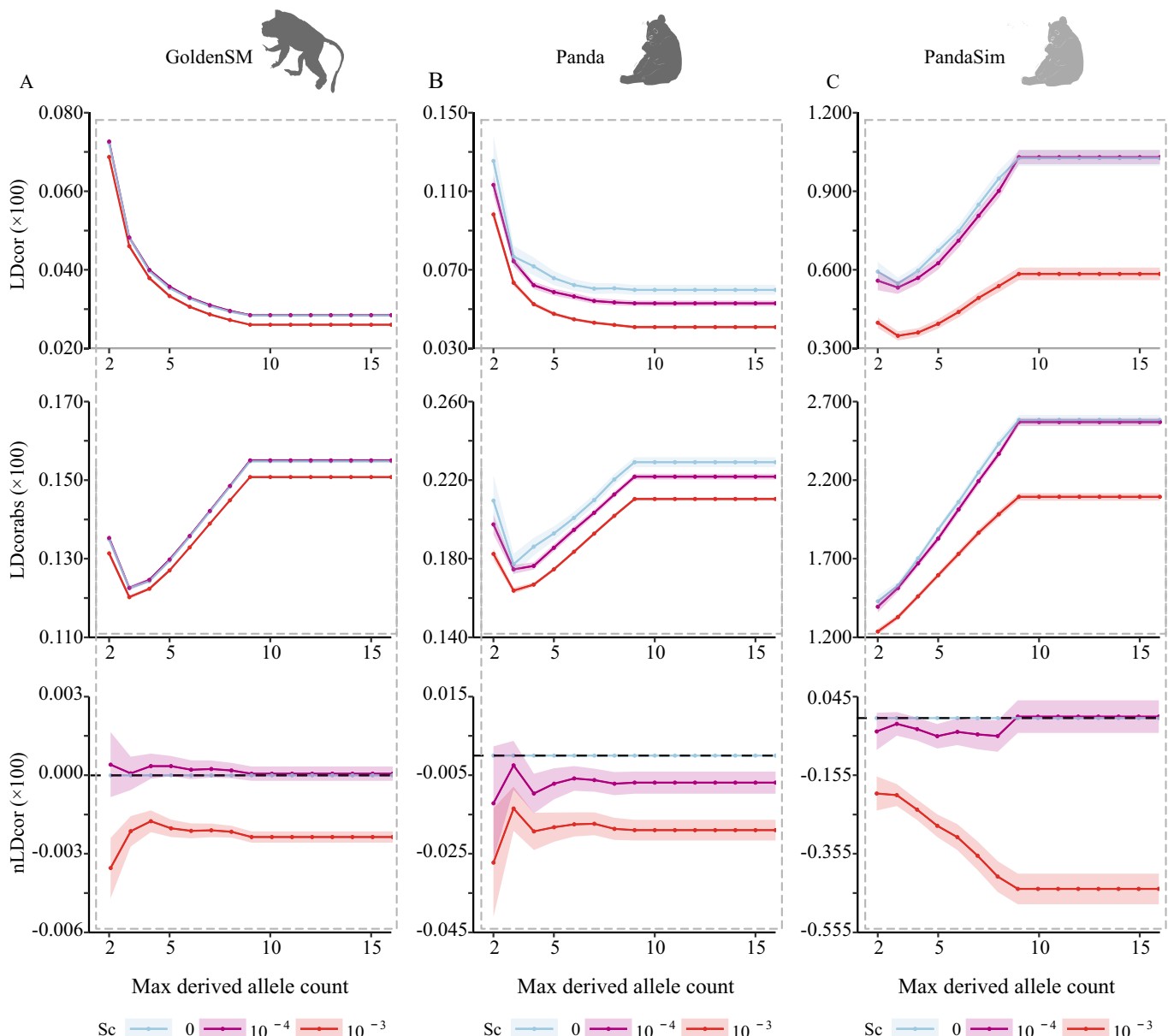

**Figure 5. Simulation results of LD with different selection coefficients and max-derived allele count in the giant panda and the golden snub-nosed monkey.**

(**A**) The demography of the golden snub-nosed monkey was simulated as inferred by Zhou et al (Zhou et al, 2016) (named as GoldenSM). The points and the error bars represent the mean value and the s.e.m. of LDcor, LDcorabs, and nLDcor. (**B**). We simulated the demography of the giant panda as inferred by Zhao et al (Zhao et al, 2013) (named as Panda). The points and the error bars represent the mean value and the s.e.m. of LDcor, LDcorabs, and nLDcor. (**C**) In addition, we scaled the population size of the giant panda to a tenfold smaller population size so as to evolve 100 generations (named as PandaSim). The points and the error bars represent the mean value and the s.e.m. of LDcor, LDcorabs, and nLDcor.

change in the direction (sign) of LD. That means that LDcorabs is consistent with the mean of $D'$ or the $\sqrt{r^2}$, with no need for genotype phasing. In addition, we have $\frac{n(n-1)}{2}\text{LDcor}_L$, which in form is very similar to the mean of signed $D'$ or signed $\sqrt{r^2}$. Furthermore, we calculate the normalized LD for deleterious loci

(nLDcor$_L$) by neutral loci as:

$$\text{nLDcor}_L = \text{LDcor}_L - \text{LDcor}_S, \tag{3}$$

$$\text{nLDcorabs}_L = \text{LDcorabs}_L - \text{LDcorabs}_S. \tag{4}$$

Where L and S are a deleterious set and a null locus set, respectively. We take the synonymous mutations as a nearly neutral reference (null) in empirical data in this paper. nLDcor and nLDcorabs decrease as negative selection coefficient gets stronger. The calculations of LDcor and LDcorabs were done in Rscript.

## The distribution of observed D for two unlinked loci

We assume a diploid random mating population of N individuals and two loci X1 (A/a) and X2 (B/b), with a derived allele count dac1 and dac2, respectively. The frequency distribution of each haplotype (AB/Ab/Ab/ab) is of finite number. The scale and possibility of each distribution and the observed D, under each distribution, are all accessible. The likelihood of each diploid distribution (diploid) is $\prod_{i,j=1}^{3} y_{ij}^{N_{ij}}$ and the combination number of that distribution is $\frac{2N!}{\prod_{i,j=1}^{3} N_{ij}!}$. The definitions are the same as described above (Weir and Cockerham, 1979). The calculation was done using Rscript.

## Forward simulation

Forward population genetics simulations were performed with SLiM v3.0 (Haller and Messer, 2019) to verify the utility of the statistics. For convenience, we began by simulating three different demographic models (Fig. 1B) that increase in complexity. Model 1 consists of a population of 10,000 individuals evolving for 100,000 generations. Model 2 is identical to the model of human demography by Gravel et al, 2011 except that there is no migration across populations. Model 3 is the model of human demography by Gravel et al, 2011. Allele frequency can be changed by natural selection (due to its effect in fitness) as well as random effect in non-infinite population (due to chance event rather than fitness, e.g., founder effect and a population bottleneck). In our simulations, we used three population sizes (i.e., 10,000, 1000, and 100) for starting size and tested each model using two scaled population size changes (scale population size to tenfold smaller and 100-fold smaller). For each simulation, we sampled 1000 individuals (100 individuals when only 100 individuals were set in the simulation) from an African population in human demography. All simulations had a length of 1 Mb and a mutation rate of $1.5 \times 10^{-8}$ per generation per base pair. The recombination rate was constant across each simulation and was fixed at $10^{-6}$, $10^{-7}$, $10^{-8}$ or $10^{-9}$ per generation per base pair. Alleles were assumed to be additive ($h = 0.5$). To test how LD changes along with the increasement of negative selection pressure and synergistic epistasis level, the strength of selection acting on deleterious alleles was set as 0, $-10^{-4}$, $-10^{-3}$, and $-10^{-2}$ and $-10^{-1}$. Synergistic epistasis acting on deleterious alleles was set as 0, $10^{-3}$, and $10^{-2}$ to simulate an increase in selection pressure. In simulations designed to explore different degrees of bottleneck intensity, as dictated by model 1's output, we adjusted the population size from the initial 10,000 to several reduced sizes: 1000, 700, and 400. We then allowed each subset to evolve across various time frames: 0, 2, 6, and 10 generations. For each unique combination of population size and duration, we conducted 120 replicates to ensure robust estimation of parameters.

## Derived allele counts influence LD

It is well-known that LD summary statistics are influenced by the allele frequency (Hill and Robertson, 2007; Eberle et al, 2006). Therefore, we controlled for this effect in the simulations by only considering pairs of single nucleotide polymorphisms (SNPs) with the same maximum derived allele frequency (e.g., MDAF). This means that all SNPs are in the same specified derived allele frequency range, in the sample. Further, we only calculated LD statistics between pairs of SNPs with the same selection coefficient.

We begin by focusing on doubletons because we hypothesized that doubletons would be enriched with polymorphisms that are more strongly influenced by negative selection relative to higher frequency variants (Cho et al, 2013; Bustamante et al, 2001; Hartl et al, 1994; Sawyer and Hartl, 1992) (Appendix Fig. S1). In addition, relative to singletons, we expected doubletons to be less influenced by sequencing errors or errors in variant calling in empirical data (Basu and Majumder, 2003). From a theoretical perspective, studying doubletons is advantageous because they are the most prevalent type of variant, after singletons, under the standard neutral model (Fu and Li, 1993).

## Empirical dataset acquisition and quality control

Population level genome sequencing data of the rhesus macaque (*Macaca mulatta*, LEAST CONCERN) (Liu et al, 2018b), the giant panda (*Ailuropoda melanoleuca*, VULNERABLE) (Zhao et al, 2013), golden snub-nosed monkey (*Rhinopithecus roxellana*, ENDANGERED) (Zhou et al, 2014), tiger (*Panthera tigris*, ENDANGERED) (Liu et al, 2018a), human (*Homo sapiens*) (The 1000 Genomes Project Consortium, 2012), sheep (*Ovis aries*) (Daetwyler et al, 2019), and gorilla (*Gorilla beringei graueri, Gorilla gorilla diehli,* and *Gorilla gorilla gorilla*, CRITICALLY ENDAN-GERED) (Prado-Martinez et al, 2013) were selected and used for subsequent study (Appendix Table S4).

For the giant panda, golden snub-nosed monkey, and tiger, reads were mapped to the respective reference genome using the Burrows-Wheeler Aligner 0.7.17-r1198 (Li, 2013), with a parameter of "-M" to mark the shorter split hits as suboptimal. Sequence Alignment/Map (SAM) format files were imported to Picard v2.0.1 (Broad Institute, 2019) for cleaning, sorting, removing duplicated reads, and converted to Binary Alignment/Map (BAM) format files. All BAM format files of a population were then imported to SAMtools v1.9 (Danecek et al, 2021) and BCFtools v1.8 (Danecek et al, 2021) to generate a pileup variant called a format (VCF) file. After that, SNPs were selected for further analysis using VCFtools v0.1.17 (Danecek et al, 2011). Only biallelic SNPs, which do not depart from Hardy-Weinberg Equilibrium ($P < 1 \times 10^{-6}$), were considered. The potential false-positive variant calls were excluded according to the following criteria: "QUAL < 20 || MQ < 40.0" in BCFtools v1.8 (Danecek et al, 2021). For each population, SNPs with DP less than 3× per sample or more than 250% of the average depth of coverage were removed as well.

We managed the genome of the macaque, gorilla, sheep, and human directly from the public VCF files. In the macaque dataset, SNPs was filtered with the criteria "-filterExpression QD < 5.0 || FS > 60.0 || MQ < 40.0 || ReadPosRankSum < -8.0 || MQRankSum < -12.5" and "−genotypeFilterExpression DP < 4.0" as the paper

advised. Only *M.m.lasiotis* samples were kept for subsequent analysis. In the gorilla dataset, SNPs were filtered as the paper mentioned if they met any of the following: "DP < (mean_read_depth/8.0) || DP > (mean_read_depth*3) || QUAL < 33 || FS > 26.0 MQ < 25", "-sites within 5 bp of a reported indelS-", or "MQ0 > = 4 && ((MQ0/(1.0*DP)) > 0.1)". In the sheep dataset, only samples of MERINO HORNED breed in Australia were kept. No filtering on coverage or genotype quality was performed on SNPs in the human data (1000 genomes populations), as the VCF files do not provide any sequencing depth (only low coverage data were counted towards the DP, exome data were not used) or quality information.

## Ancestral allele reconstruction

We established the ancestral allele by leveraging the genome sequences of targeted species and their closely related counterparts (Appendix Table S5). For each species, a one-to-one genome alignment with its closely related species was produced by LAST v2.31.1 (Hamada et al, 2017; Kiełbasa et al, 2011), following the parameters and commands used in generating "2017 human-ape alignments". Subsequently, for biallelic SNPs that were fully represented in the VCF file (i.e., without missing sites), we extracted precise coordinate conversion details using MafFilter v1.3.1 (Dutheil et al, 2014). When the reference allele of the related reference species corresponded to either the reference or alternative allele of the target species, it was designated as the ancestral allele. For the CEU human population, sheep population and additional gorilla samples, we utilized chain files from UCSC and applied LiftOver (Kuhn et al, 2013) or direct coordinate conversion information retrieval.

## Variant annotation

Functional consequences of genetic variants were annotated using Ensemble Variant Effect Predictor version 0.1.13 (McLaren et al, 2016). Alleles were classified as synonymous, missense (nonsynonymous), stop gained mutations (resulting in a premature stop codon leading to a shortened transcript), stop lost mutations (where at least one base of the terminator stop codon has changed resulting in an elongated transcript), splice acceptor variant mutations (that change the 2-base region at the 3' end of an intron), and splice donor variant mutations (that change the 2-base region at the 5' end of an intron). Loss-of-function (LoF) alleles were defined as the joint set of "stop gained", "stop lost", "splice acceptor variant" and "splice donor variant" alleles (Appendix Table S6). For the empirical data, we divided the filtered SNPs into three groups according to the annotation: synonymous, missense, and LoF. For each of the three SNPs sets, we calculated the LD statistics as indicated above.

## Resampling and permutation tests

In order to preclude the possible bias caused by different derived allele frequency spectrums, we resampled variants in the neutral loci set to get 100 new neutral sets, each with the same DAF of the non-neutral variants set. The p values of the LD statistics in LoF sets were accessible during this process. We calculated LD for each run and then applied a permutation test. Analogously, we resampled the same derived allele frequency spectrum of LoF loci from synonymous loci and missense loci across 100 runs in the empirical data and applied permutation tests as well.

## Data availability

The datasets used in in this study are all publicly available as listed in Appendix Table S4. The source data of this paper are available at https://github.com/Chunyan-Hu/LDcor.git. Any additional information required to reanalyze the data reported in this paper is available from the lead contact author upon request.

The source data of this paper are collected in the following database record: biostudies:S-SCDT-10_1038-S44319-024-00307-2.

## Peer review information

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

## Acknowledgements

The authors thank Mashaal Sohail for sharing the R code. This project was granted by the National Key Research and Development Projects of the Ministry of Science and Technology of China (2023YFC3304000), the National Natural Science Foundation of China (32270437), and the Institute of Zoology, Chinese Academy of Sciences (2023IOZ0104). PAG wishes to thank Chrissie, Sara, Jenni, Dax, and Saffron for their love and support.

## Author contributions

**Chunyan Hu**: Data curation; Formal analysis; Validation; Visualization; Writing—original draft; Writing—review and editing. **Gaoming Liu**: Data curation; Writing—original draft. **Zhan Zhang**: Data curation; Visualization. **Qi Pan**: Data curation; Formal analysis. **Xiaoxiao Zhang**: Resources; Data curation. **Weiqiang Liu**: Methodology; Writing—original draft. **Zihao Li**: Resources; Visualization. **Meng Li**: Writing—original draft; Project administration. **Pingfen Zhu**: Validation; Writing—review and editing. **Ting Ji**: Methodology; Writing—original draft; Writing—review and editing. **Paul A Garber**: Supervision; Validation; Writing—original draft; Writing—review and editing. **Xuming Zhou**: Conceptualization; Supervision; Funding acquisition; Investigation; Visualization; Methodology; Writing—original draft; Project administration; Writing—review and editing.

Source data underlying figure panels in this paper may have individual authorship assigned. Where available, figure panel/source data authorship is listed in the following database record: biostudies:S-SCDT-10_1038-S44319-024-00307-2.

## Disclosure and competing interests statement

The authors declare no competing interests.

