## [Peer Review File · EMBO Reports]

Genetic linkage disequilibrium of deleterious mutations in threatened mammals

Chunyan Hu, Gaoming Liu, Zhan Zhang, Qi Pan, Xiaoxiao Zhang, Weiqiang Liu, Zihao Li, Meng Li, Pingfen Zhu, Ting Ji, Paul Garber, and Xuming Zhou

Corresponding author(s): Xuming Zhou (zhouxuming@ioz.ac.cn)

Review Timeline:

Submission Date:	28th Sep 23
Editorial Decision:	31st Jan 24
Revision Received:	23rd Apr 24
Editorial Decision:	16th Jul 24
Revision Received:	26th Aug 24
Editorial Decision:	8th Oct 24
Revision Received:	9th Oct 24
Accepted:	16th Oct 24

Editor: Yehu Moran

Transaction Report:

Dear Dr. Zhou

First, I would like to apologize for the unusual delay in sending to you this decision letter. It proved challenging to find three expert Referees but finally we managed and I believe their reports can help you to improve your manuscript.

As you will see, the referees acknowledge that the findings are potentially interesting and can significantly contribute to the field of conservation genomics. However, they also raise significant concerns that should be addressed.

I would thus like to invite you to revise your manuscript with the understanding that the referee concerns must be fully addressed and their suggestions taken on board. Please address all referee concerns in a complete point-by-point response. Acceptance of the manuscript will depend on a positive outcome of a second round of review. It is EMBO Reports' policy to allow a single round of major revision only and acceptance or rejection of the manuscript will therefore depend on the completeness of your responses included in the next, final version of the manuscript.

We realize that it is difficult to revise to a specific deadline. In the interest of protecting the conceptual advance provided by the work, we recommend a revision within 3 months (2nd May 2024). Please discuss the revision progress ahead of this time with the editor if you require more time to complete the revisions.

NOTE from the Editor: Please put special attention to how you divide your text between Results and Discussion. My impression is that some of the more hypothetical points in your results might fit better in the Discussion. Moreover, please try to avoid any unnecessary "hyperbolic" claims that are not necessary or not well-supported by the results.

- 1) A data availability section providing access to data deposited in public databases is missing. If you have not deposited any data, please add a sentence to the data availability section that explains that.
- 2) Your manuscript contains statistics and error bars based on $n=2$. Please use scatter blots in these cases. No statistics should be calculated if $n=2$.

3) We replaced Supplementary Information with Expanded View (EV) Figures and Tables that are collapsible/expandable online. A maximum of 5 EV Figures can be typeset. EV Figures should be cited as 'Figure EV1, Figure EV2' etc... in the text and their respective legends should be included in the main text after the legends of regular figures.

5) a complete author checklist, which you can download from our author guidelines <https://www.embopress.org/page/journal/14693178/authorguide>. Please insert information in the checklist that is also reflected in the manuscript. The completed author checklist will also be part of the RPF.

6) Please note that all corresponding authors are required to supply an ORCID ID for their name upon submission of a revised manuscript (<https://orcid.org/>). Please find instructions on how to link your ORCID ID to your account in our manuscript tracking system in our Author guidelines

<<https://www.embopress.org/page/journal/14693178/authorguide#authorshippingguidelines>>

I look forward to seeing a revised form of your manuscript when it is ready.

Yours sincerely,

Yehu Moran
Academic Editor
EMBO Reports

Referee #1:

"Deleterious mutations are not commonly overdispersed in threatened mammals" by Hu et al introduces LD statistics for unphased genetic data and uses them to explore the LD characteristics of six mammalian species. They use model-based simulations to explore the characteristics of their proposed statistics under a few demographic models, and then turn to empirical data. I appreciate several aspects of the manuscript -- characterizing how evolutionary processes shape genomic variation in threatened species is an extremely important problem, and I admire the authors' approach of using model-based simulations to guide their interpretation. There are some aspects of the manuscript that, in my opinion, could be substantially revised to increase its potential impact and readability.

Major comments:

The manuscript begins by outlining two LD statistics that can be calculated from unphased genetic data and highlights some similarities to the classic statistics r^2 and D' . While the authors do compare to one other LD statistic that was developed for unphased data (NetLD), there has been quite a lot of other work published on LD statistics for unphased data. Since this work isn't discussed much in the manuscript it's a bit hard to tell whether the authors' statistics may provide different information than previous methods for estimating LD from unphased data. For example, the statistic σ^2_d defined in Ragsdale & Gravel ("Unbiased Estimation of Linkage Disequilibrium from Unphased Data") is similar in form, and was proposed in earlier studies. Some LD statistics (e.g. D^2) can be estimated from unphased data using the EM algorithm (e.g., Slatkin and Excoffier "Testing for linkage disequilibrium in genotypic data using the Expectation-Maximization algorithm", or Rogers "How Population Growth Affects Linkage Disequilibrium").

A richer discussion of these existing methods/statistics to motivate if/why the new statistics are needed and what advantages (either practical or theoretical) over existing approaches would help motivate the study much better. Alternatively, description of the statistics could be limited to a shorter section simply describing how they capture the essence of the original D' and r^2 .

The authors then explore how their statistics vary with demography, selection strength, and allele frequency. I very much appreciate how the authors used model-based simulations here. They focus primarily on models related to human demography. The main goal of the study seems to be to link LD patterns at putatively deleterious variants to conservation status. If the statistics can really be useful in interpreting the effects of evolutionary processes on conservation status, it would be helpful to consider a much broader range of demographic models that correspond to bottlenecks of varying magnitude, as have been experienced by many threatened species. As it stands, it is not clear how to relate the simulation findings to conservation status or evolutionary processes that may be related to declining population sizes.

I think the title of the paper extends quite a bit past what can be gleaned from the LD stats the authors propose. There are only three species of conservation concern in the analysis. The LD patterns are far from being clear across these species, and do not seem straightforward to relate to the simulations presented in the paper.

The methods could use quite a lot of clarification. What is the definition of "maximum derived allele frequency"? A single allele only has single corresponding allele frequency based on its representation in the sample, so I am unclear on what is being maximized. Similarly, what is being permuted in the permutation test? What is the model of epistasis that was used, and how does this map to the specific model parameters? Some additional care to explain model assumptions and/or the detail of calculations throughout the methods would be very helpful for readers. Many of the plots were difficult for me to interpret because I did not fully understand these details based on the manuscript.

In sum, I think this is a potentially very interesting idea that could be impactful. Aspects of both the presentation and analyses in the manuscript could be substantially revised to increase the potential impact of this research.

Referee #2:

Comments to the Author

Zhou et al. introduce in this manuscript an interesting new set of statistics to estimate mutational load based in linkage disequilibrium among deleterious loci. The theoretical background is solid, the Methods section is well explained and the mathematical approach rather robust, simulations are correctly run and the examples are (in principle) correctly chosen. However, I have some concerns about the structure of the paper, especially after the section "The new LD statistics have great

consistency...". Some claims sparkled in the Results section do not have enough evidence to be sustained with the present analyses or look like they would fit better in Discussion. Some implications of the results of the paper cannot be drawn with the evidence presented. Also, most figures and supplementary figures have footers which are not well worded and would need to be expressed more clearly. All in all, I think that this manuscript brings a couple new set of statistics that would be of great use for the scientific community that works in related fields; however, it needs a deep revision of some sections and some re-thinking of the theoretical framework. Comments below.

Comments about text:

181 - "and was implemented into SliM v3.0" this is redundant to the previous mention of SliM and does not need to be here.

183 - "For each simulation, we sampled 1,000 individuals (100 individuals when only 100 individuals were set in the simulation) from the African population in the human demography."

This means that for the simulation with 10,000 individuals, 10% of them are sampled as subset. This can greatly increase the speed of calculation of statistics, but there are an increase of biases and false positives due to Wahlund effect in some loci. In my personal experience, it is better to test if subsetting the population to 10% is going to affect the estimation of statistics before proceeding. I would recommend, for the sake of time, to perform such test with the population of 1,000 by sampling 100 individuals and seeing if the estimations are consistent to what you found by sampling 1,000 out of 1,000.

224 - "For each population, SNPs with DP in the upper 5% (giant panda:426, rhesus macaque:302, golden snub-nosed monkey: 786, tiger: 531, and sheep: 412) or lower 5% (giant panda: 200, rhesus macaque:4, golden snub-nosed monkey: 181, tiger: 197, and sheep: 204)" - can the authors provide a reason why they have followed this criterion? Normally the DP filter is set up at a minimum of 3-5X per sample and 200% (or 250%) of the average depth of coverage.

285-289 "The LD pattern of recessive deleterious mutations is due to the complex interplay of heterosis effect (associative overdominance) whereby these mutations are masked from selection in the heterozygous state. This effect may be exaggerated if there is a small degree of recombination and the deleterious alleles arise on the same haplotype, leading to positive Ldcor." This sounds like a precipitate conclusion and the connection to the results is not clear. I would appreciate if authors could elaborate more about why this is the case and cite a similar case from the reference 35, stating what is the connection between this conclusion and the reference. Also, this would fit better in Discussion.

296 - 301 "For any selection coefficient, Ldcor always decreases monotonically as the synergistic epistasis level increases. Under the same synergistic epistasis level, Ldcor usually decreases monotonically as the negative selection coefficient increases. Both negative selection and synergistic epistasis can increase negative LD, suggesting that an individual locus may not be the appropriate unit of selection"

Figure 2 is not appropriately addressed in this section (until line 321). If such a conclusion "Both negative selection and..." is to be drawn, it should be at the end of the entire section, or better subdivide Fig. 2 into 3 sections (A, B, C) and correctly point out to each sub-figure with each logical conclusion and explanation of results per separate. The text is not very solid, since after this last statement we have "We next examined... (line 302)". I would suggest redoing this section, addressing the subfigures of Figure 2 one by one.

303 "In simulations with 10,000 individuals evolving" - delete evolving.

305 - 308: "For any specific selection coefficient, model 1 had the largest Ldcor value, followed by model 3 and model 2, suggesting that population expansion may produce less LD while increased gene flow contributes to increases in LD". This is not what can be seen in Fig. 2A (left). Model 3, for $s = -10^{-4}$, it looks like it has a higher value than model 1. If this is not the case, since the graph is not entirely clear I would suggest authors to mention the numerical values and explain how higher they are to each other.

308: "Genetic drift diminished greatly after population expansion..." This is the first reference to genetic drift in the entire manuscript and there is no way of the authors to detect it. This argument is not solidly presented.

309-310: "In model 3 both gene flow and differences in allele frequency among subpopulations contributed to LD" Although I understand the meaning of this sentence and why the authors mention it, this feels like it does not fit in this paragraph. This is not a directly viewable result and I would suggest moving it to Discussion.

335 - "While, LD" - Perhaps you mean "Meanwhile, LD..."?

336 - "when loci far located" - I would add "to each other". Also, it looks like Figures S6 and S7 and all text associated to them should be moved to section "Estimation of LD in six mammalian species".

355 - 364. This paragraph should be in Discussion.

376 "while Ldcor was always decreased" it should be "while Ldcor always decreased"

389 - 391: "Three of these are recognized by the IUCN as threatened species: the goldenSM, tiger, and giant panda." Aesthetically, it would be better to include this information within the previous sentence or in Methods, not here.

394 - 400 "In our results..." There is no reference to any figure or table. Where are these results extracted from?

401 - 403 "For most of the nondomesticated populations, the distribution of rare (e.g., derive allele count ≤ 5) LoF allele sets were underdispersed relative to synonymous and missense sets of the same derived allele count" I don't see clear how this conclusion is extracted from Table 2 and Fig. 4.

404 - "This pattern was consistent even when allele frequency was controlled (Fig. S8)" Perhaps you mean Fig. S6?

410 - 411 "similar situation was observed in human, goldenSM (Endangered), and the giant panda (Vulnerable)." However, LoF Ldcor values are higher than synonymous and missense in goldenSM above DAF > 5 . I would suggest adding the DAF scale to each of the subgraphs of Ldcor and Ldcorabs, since it is not easy to see.

The whole paragraph (405 - 420) should address properly and clearly between which values of DAF these trends of Ldcor are higher or lower for LoF.

429 "suggesting that giant panda" either "suggesting that giant pandas" or "suggesting that the giant panda"

432 - 435 "We further scaled the population size of giant pandas to 10-fold smaller and then evolved 100 generations. The results indicate that the LD of slightly deleterious variants was mixed with that of neutral variants (Fig. 5), suggesting an increase of potential genetic risk." This paragraph is terribly worded and obscure. A different way to write it would be: "We further scaled the population size of giant pandas down 10-fold in our simulation and ran it for 100 generations. The results indicate that the LD values/trend of slightly deleterious variants was indistinguishable from that of neutral variants". The explanation of increase of genetic risk is not appropriately done.

441 "ignores the linkage among different loci" better "ignored the importance of linkage..."

444 "ignores the direction". Of what? Perhaps of linkage disequilibrium?

452 "V/Va" This is the first mention to this equation in the entire paper.

490 - 496 "We tested... are small (Fig. S10)" This entire section belongs to Results.

499 - 500 "explain how humans have accommodated high mutation rates over our evolutionary history." I would need a reference for this ascertainment.

528 "Both the two" is redundant.

I would change "table" for "Table" in all references in text.

Fig. 2 "(A) We simulated different synergistic epistasis levels (0, 10⁻¹, 10⁻², 10⁻³, and 10⁻⁴) as well as different negative selective coefficients (0, 10⁻², 10⁻³, and 10⁻⁴) for each model (10,000 individuals) with a recombination rate of 1×10^{-8} crossovers per base pair per generation (left). The effect of different models and demography (Sample 1 (10,000, 1,000): with 10,000 individuals evolving and 1,000 individuals used in the calculation; Sample 2 (1,000, 1,000): with 1,000 individuals evolving and 1,000 individuals used in the calculation; Sample 3 (100, 100): with 100 individuals evolving and 100 individuals used in the calculation) on Ldcor for different selection coefficients (right)" This figure is not appropriately explained. Both the organization of the image and the footer are confusing, since (A) left is depicting something entirely different to (A) right. I would subdivide the figure in 3 (A, B, C). Also, there is no reference to what each line means, neither at the footer nor in the image.

Fig. 3 "The square root of classical r^2 (signed by D)" - you should find a better way to express this statistic instead of a piece of text. Perhaps "signed". The colors of this figure are difficult to read, we would need them to be more intense. Also, I would suggest using a colorblind-friendly palette. Also, I would eliminate the explanatory text "Ldcor and Ldcorabs show great...". This would fit in the main text, not here.

Fig. 4 "codingsynononly" is a mistake. It should be "synonymous"

Fig. 5 "Careful attention must be paid to the population genomic health of golden snub-nosed monkey (left)." What is "population genomic health"?

Fig. S2 needs more resolution. It is hard to see differences between the dark dotted line with dots and the line with "X". I would also change the name of "0" simulations and "-0.01" simulations. Perhaps naming the datasets would help mention them more easily.

Fig. S3 "(A) For all three models (model1: constant population size; model 3: with population expansion and gene flow; model 2: with population expansion), we did simulations 120 runs for each recombination rate and each selection coefficient. The point and the error bar represent mean value of Ldcor and a standard error mean. (B) For all three models, we did simulations 120 runs for each recombination rate and each selection coefficient. The point and the error bar represent mean value of Ldcorabs and a standard error mean."

These are not appropriate explanations for this figure. A better way of expressing what is in the figure would be: "(A) Ldcor values in three models (model1: ; model3: ...). We ran 120 simulations for each recombination rate and each selection coefficient. The point and the error bar..."

Fig. S4 - same as with Fig. 2, it should be divided in 3 sections and its results should be more precisely addressed in main text.

Fig. S5 "with 10000 individuals evolving and 1000 individuals in calculating" This is not a good way to express results. I would say, "a simulation with 10000 individuals and a subset of 1000 individuals"

Fig. S8 - A notation of only 2 decimals would be better in the third set of graphs (right)

Referee #3:

The authors introduce new statistics to detect and compare global linkage disequilibrium of deleterious mutations across species, using unphased genotypes. The study employs simulation analyses to validate the effectiveness of the new statistics in standardizing differences among mutations with varying allele frequencies. The analysis focuses on six mammal species, namely sheep, tigers, golden snub-nosed monkeys, giant pandas, humans, and rhesus macaques. The results reveal overdispersion of deleterious mutations in sheep, tigers, and golden snub-nosed monkeys, suggesting less effective negative selection. Conversely, strong negative linkage disequilibrium of deleterious mutations is observed in giant pandas, humans, and rhesus macaques. The introduction of new statistics to assess global linkage disequilibrium is a notable contribution, providing a novel approach for evaluating selection intensity. However, some aspects should be clarified for a more comprehensive understanding.

My specific comments include:

- 1) The rationale for choosing the six specific mammal species and the implications of their observed selection patterns should be elaborated. Why the selection of the species studied (all unrelated)? It would be interesting to see also a comparison between some closely related species using shared (ancestral) loci/genes, such as human and other close ape species, for example. It could give more intuitive insights combined with clear known evolutionary path.
- 2) The normalization of LD based on synonymous mutations (l. 157-159, l. 472), would require some further elaboration. Could this be introducing some bias by assuming that synonymous mutations are always neutral and homogeneous. For instance, currently selection tests already assume the variability of synonymous mutation rates. Would it be also relevant here to take it in consideration?
- 3) In the methods "ancestral genome reconstruction" (l. 235) it is mentioned closely related species, but no further detail provided (what species, divergence times, etc...).
- 4) The term "narrowing purified selection" might benefit from further clarification to enhance reader comprehension.
- 5) Consider introducing the two new statistical approaches, namely the averaged pairwise standardized covariance (LDcor) and the averaged absolute pairwise standardized covariance (LDcorabs), in the abstract for a more comprehensive overview.
- 6) Maintain consistency by either using "purifying" or "negative" selection throughout the manuscript.
- 7) Before comparing with your statistical methods, introduce both classical r^2 and D' (signed by D) and NetLD.
- 8) To facilitate readers, especially those interested in employing your statistical methods, consider providing the R script mentioned in the manuscript. This would enhance transparency and reproducibility.
- 9) Overall the English needs improvements. There are many small mistakes and grammatical errors throughout the manuscript that would need revision.

Detailed responses to editors and reviewers

All comments provided by Editor and reviewers are shown in gray italics, and our responses are shown in blue. All revisions in the manuscript are marked in red.

EDITOR COMMENTS

Dear Dr. Zhou

I would thus like to invite you to revise your manuscript with the understanding that the referee concerns must be fully addressed and their suggestions taken on board.

Please address all referee concerns in a complete point-by-point response.

Acceptance of the manuscript will depend on a positive outcome of a second round of review. It is EMBO Reports' policy to allow a single round of major revision only and acceptance or rejection of the manuscript will therefore depend on the completeness of your responses included in the next, final version of the manuscript.

We realize that it is difficult to revise to a specific deadline. In the interest of protecting the conceptual advance provided by the work, we recommend a revision within 3 months (2nd May 2024). Please discuss the revision progress ahead of this time with the editor if you require more time to complete the revisions.

NOTE from the Editor: Please put special attention to how you divide your text between Results and Discussion. My impression is that some of the more hypothetical points in your results might fit better in the Discussion. Moreover, please try to avoid any unnecessary "hyperbolic" claims that are not necessary or not well-supported by the results.

◦ ◦ ◦

I look forward to seeing a revised form of your manuscript when it is ready. Please use this link to submit your revision: <https://embor.msubmit.net/cgi-bin/main.plex>

Response:

Thank you very much for your invaluable support and constructive feedback on our manuscript. We deeply appreciate the comprehensive suggestions brought forward by the referees, which span several key areas: the reorganization of Figure 2 and its clearer demarcation between the Results and Discussion sections, enriching the manuscript with simulations depicting various bottleneck scenarios, enhancing it with empirical datasets from closely related species, clarifying specific definitions more thoroughly, and fine-tuning language nuances. The primary areas of focus included the reconfiguration of Figure 2 (addressed in Comments 1-4, 2-5, 2-7, 2-29, 2-35), the appropriate allocation of critical statements within either the Results or Discussion sections (addressed in Comments 2-4, 2-5, 2-9, 2-12), alongside new content elucidating the implications of reducing the population sample to 10% (addressed in Comment 2-2), and the influence of bottleneck events on our statistical findings (addressed in Comment 1-2).

Specifically, to delineate the differential impacts of synergy levels and demographic factors on Linkage Disequilibrium (LD), we have now represented these aspects in distinct figures: Figure 2A (formerly the left part of Fig. 2A) and Figure 2C (formerly the right part of Fig. 2A). To explore the effects of subsampling the population to 10% and the impacts of bottleneck phenomena on our statistics, we introduced Figure 2B, which was previously not included in Figure 2. Additionally, Figure 2D (formerly Figure 2B) now presents the influence of loci distance on LD. These adjustments and the respective discussions on LD's interaction with the tested models are now clearly sectioned within the "Results" (lines 384-471). We have also restructured the division between the Results and Discussion sections in accordance with the referees' recommendations.

Furthermore, we have updated the figures to include a colorblind-friendly palette and meticulously reviewed the manuscript language to rectify any grammatical inaccuracies. The corresponding scripts have been made available at (<https://github.com/Chunyan-Hu/LDcor.git>) for further reference and transparency.

REFEREES COMMENTS

Referee #1

"Deleterious mutations are not commonly overdispersed in threatened mammals" by Hu et al introduces LD statistics for unphased genetic data and uses them to explore the LD characteristics of six mammalian species. They use model-based simulations to explore the characteristics of their proposed statistics under a few demographic models, and then turn to empirical data. I appreciate several aspects of the manuscript -- characterizing how evolutionary processes shape genomic variation in threatened species is an extremely important problem, and I admire the authors' approach of using model-based simulations to guide their interpretation. There are some aspects of the manuscript that, in my opinion, could be substantially revised to increase its potential impact and readability.

Response:

We sincerely appreciate your efforts in summarizing our study and providing valuable comments. Your acknowledgment of our work, especially the emphasis on integrating model-based simulations and empirical data to demonstrate the utility of LDcor and LDcorabs, is highly encouraging. In the updated version, we have taken steps to enhance the visual appeal of our figures through color reorganization and improved the manuscript's structure for better readability. Below, you will find a detailed response to each point raised. We trust that our revisions accurately capture the essence of your inquiries and effectively address any concerns you might have.

Comment 1-1:

The manuscript begins by outlining two LD statistics that can be calculated from unphased genetic data and highlights some similarities to the classic statistics r^2 and D' . While the authors do compare to one other LD statistic that was developed for unphased data (NetLD), there has been quite a lot of other work published on LD statistics for unphased data. Since this work isn't discussed much in the manuscript it's a bit hard to tell whether the authors' statistics may provide different information than previous methods for estimating LD from unphased data. For example, the statistic σ^2_d defined in Ragsdale & Gravel ("Unbiased Estimation of Linkage Disequilibrium from Unphased Data") is similar in form, and was proposed in earlier studies. Some LD statistics (e.g. D^2) can be estimated from unphased data using the EM algorithm (e.g., Slatkin and Excoffier "Testing for linkage disequilibrium in genotypic data using the Expectation-Maximization algorithm", or Rogers "How Population Growth Affects Linkage Disequilibrium").

A richer discussion of these existing methods/statistics to motivate if/why the new statistics are needed and what advantages (either practical or theoretical) over existing approaches would help motivate the study much better. Alternatively, description of the statistics could be limited to a shorter section simply describing

how they capture the essence of the original D' and r^2 .

Response:

Thanks for the suggestion. We appreciate your feedback. In our revised manuscript, we've expanded our comparison to include additional statistics for unphased data, which should demonstrate the improvements of our metrics. We add this information to the “Discussion” in our new version (lines 472-494):

“A classical LD-detecting statistic, D , which depends on allele frequency and ignores the direction of linkage disequilibrium, measures the difference between observed and expected frequencies of a haplotype. The more popular statistics D' and r^2 represent standardized D , and therefore are not limited by frequency. However, they still obscure the direction of linkage disequilibrium⁵². Moreover, the classical r^2 and D' depend on haplotype information, which is often difficult to obtain. This challenge has steered recent research towards analyses based on unphased data. This body of work generally falls into two approaches: one involves estimating haplotype information to then calculate traditional or modified traditional statistics (e.g., σ_d^2 ⁵³), whereas the other bypasses the need for haplotypes entirely (for example, NetLD²³, and $H_R^{(j)}$ ¹⁹). However, these methods come with limitations. First, estimations, particularly those relying on EM iterations, may not always pinpoint the global maximum, leading to potential inaccuracies⁵⁴. Second, these computations are not only resource-intensive but also become more complex with the inclusion of additional SNPs, with haplotypes comprising three SNPs being more variable than those with two⁵³. Third, for two SNPs located on different chromosomes or contigs, such estimation processes fail to yield actual haplotypes. The recent developed $H_R^{(j)}$ ¹⁹ can estimate LD by averaging counts of doubly heterozygous (0/1,0/1) for all pair of SNPs in a set, however, it ignores all other genotype information without normalization. Similarly, NetLD²³ averages covariance across loci pairs, overlooking variance differences between them and estimating LD without accounting for discrepancies in derived allele frequencies, thus lacking comparability.”

Comment 1-2:

The authors then explore how their statistics vary with demography, selection strength, and allele frequency. I very much appreciate how the authors used model-based simulations here. They focus primarily on models related to human demography. The main goal of the study seems to be to link LD patterns at putatively deleterious variants to conservation status. If the statistics can really be useful in interpreting the effects of evolutionary processes on conservation status, it would be helpful to consider a much broader range of demographic models that correspond to bottlenecks of varying magnitude, as have been experienced by many threatened species. As it stands, it is not clear how to relate the simulation findings to conservation status or evolutionary processes that may be related to declining population sizes.

Response:

Thank you for the valuable suggestion. In our previous iteration, we acknowledged the challenges posed by the small population sizes and diverse demographics characteristic of many endangered species. We specifically modeled how population size impacts linkage disequilibrium (LD). Our direct comparison of the effects of population scaling—from 10,000 to 1,000 and down to 100—revealed that both the magnitude and variability of LD_{cor} increased with decreasing population size for any given selection coefficient (as illustrated in Fig. R1). This pattern emerged consistently across three different population sizes (10,000, 1,000, and 100) and three models. Notably, we observed a monotonic decrease in LD_{cor} as the negative selection coefficient intensified across all models. This trend leads us to conclude that our statistical approach is effectively responsive to variations within small population contexts.

However, we agree with your opinion that declining in population size matters as well. During revision, we have incorporated simulations on three different scale sizes (with a decline step of 300), as well as various durations of bottlenecks (0, 2, 6, and 10 generations). In order to clearly indicate the effect of decline, simulation was done under the Model 1 (with constant population size and evolving for 100,000 generations). After the 100,000 generations evolving for 10000 individuals, we sampled subsets with 1000, 700, and 400 individuals separately, and let each of the subset to evolve for four different durations: 0, 2, 6, and 10 generations. All simulations had a length of 1 Mb and a mutation rate of 1.5×10^{-8} per generation per base pair. The recombination rate was constant across each simulation and was fixed at 1×10^{-8} per generation per base pair. Alleles were assumed to be additive ($h = 0.5$). To test how LD changes along with the increasement of negative selection pressure, the strength of selection acting on deleterious alleles was set as 0, -10^{-4} , -10^{-3} , and -10^{-2} and -10^{-1} . Each parameters set was estimated with at least 120 replicates.

Our findings indicate that as the negative selection intensity increases, the level of LD generally decreases for a given scale size or duration. Interestingly, we have observed that occasionally, the LD of nearly neutral variants ($s=-0.0001$) surpasses that of neutral variants ($s=0$). Furthermore, our results demonstrate that the LD tends to increase as the scaling size decreases, irrespective of the tested duration. Besides, for any given scaling size, the LD typically increases with the duration. In short, threatened species that related to declining population size is preferred to be with higher LD than other species.

These findings strength that the LD can effectively represent the selection intensity across diverse population decline scenarios. We have incorporated this information into Figure 2 of our study (Fig. 2B).

Fig. R1: LDcor value under varying population size. For all three tested models (model 1: demography with constant population size; model 2: demography with population expansion; model 3: demography with population expansion and gene flow), we simulated and compared LDcor under different population sizes (1000, 1000, and 100).

Fig. 2B: In order to compare the LDcor of varying magnitudes of bottleneck (based on model 1), we simulated different scaled population size (from 10000 to 1000: bottleneck 1, 700: bottleneck2, and 400: bottleneck 3) as well as different duration time (0, 2, 6, and 10 generations). And the dark gray line is the results of no-resampling (10000) at time 0 (e.g., time 00). Fig. 2C The effect of different models and demography on LDcor for different selection coefficients (sample1 (10000,1000): a simulation with 10000 individuals and a subset of 1000 individuals in calculating; sample2 (1000,1000): a simulation with 1000 individuals and a subset of 1000 individuals in calculating; sample3 (100,100): a simulation with 100 individuals and a subset of 100 individuals in calculating).

Comment 1-3:

I think the title of the paper extends quite a bit past what can be gleaned from the LD stats the authors propose. There are only three species of conservation concern in the analysis. The LD patterns are far from being clear across these species, and do not seem straightforward to relate to the simulations presented in the paper.

Response:

Your advice of limiting the title to a smaller area is pertinent. Limited number of tested species varying a few clades is not representative enough. We are thinking to use “Developing a new measurement of linkage disequilibrium between deleterious mutations and its application in threatened mammals”. However, there is limitation of characters in title, we now changed our title as “Measurement of linkage disequilibrium between deleterious mutations in threatened mammals”.

Comment 1-4:

The methods could use quite a lot of clarification. What is the definition of "maximum derived allele frequency"? A single allele only has single corresponding allele frequency based on its representation in the sample, so I am unclear on what is being maximized. Similarly, what is being permuted in the permutation test? What is the model of epistasis that was used, and how does this map to the specific model parameters? Some additional care to explain model assumptions and/or the detail of calculations throughout the methods would be very helpful for readers. Many of the plots were difficult for me to interpret because I did not fully understand these details based on the manuscript.

Response:

Apologies for any confusion caused. The statistics in question are not applicable to individual SNP but rather to a selected group of SNPs, such as a set of loss-of-function (LoF) loci or a synonymous loci set within a given population. The term "maximum derived allele frequency" (MDAF) refers to the highest frequency observed for the derived allele—a variant that emerged after the most recent common ancestor—in the SNP set under consideration. Hence, within this group, all SNPs will display a frequency of the derived allele that is either equal to or less than the defined MDAF value (for instance, an MDAF of 0.01 implies that the frequencies of the derived alleles fall within the range of (0, 0.01]).

The permutation testing is carried out for comparing two subsets of SNPs, like LoF and synonymous loci sets, within the population, aiming to mitigate potential biases that originate from varied derived allele frequency (DAF) spectra between the two sets. This permutation approach involves randomly resampling the SNPs from the synonymous set to produce numerous new synonymous sets, each with a DAF spectrum that matches that of the LoF set they are being compared with.

To assess how different levels of epistasis impact linkage disequilibrium (LD) across a range of models and selection intensities, our simulations incorporated a spectrum of epistasis levels, specifically 0, 10^{-3} , and 10^{-2} , for all the parameters being tested. Figure 2A is composed of three sub-figures, each illustrating the outcomes related to epistasis under models 1 through 3. Take, for example, the first sub-figure within Figure 2A, which represents the findings under model 1: here, the horizontal axis delineates the range of selection coefficients being scrutinized, and the vertical axis delineates the range of selection coefficients being scrutinized, and the vertical axis denotes the corresponding values of LD correlation (LDcor) detected. A variety of colors are utilized to distinguish amongst the different levels of epistasis represented.

A Simulations under different epistasis levels (left) and different models (right)

Fig. 2: LDcor with different levels of synergistic epistasis, bottleneck, demography, and maximum derived allele frequency.

Comment 1-5:

In sum, I think this is a potentially very interesting idea that could be impactful. Aspects of both the presentation and analyses in the manuscript could be substantially revised to increase the potential impact of this research.

Response:

We are profoundly grateful for the constructive suggestion you've offered. The insights you shared regarding the title (Comment 1-3), the comparative analysis of theoretical frameworks (Comment 1-1), the design of our simulations (Comment 1-2), and the overall presentation of our manuscript (Comment 1-4) have been extraordinarily valuable, instigating a thorough reassessment on our end.

In response to your feedback, we opted for a more reserved title in the revised version: "Measurement of linkage disequilibrium between deleterious mutations in threatened mammals". This change underscores our commitment to a nuanced exploration of the topic. Furthermore, we have enriched the discussion by incorporating a more detailed comparison of existing LD statistics, which lays the groundwork for highlighting the advantages offered by our novel statistical approach. Additionally, we have expanded our simulations to include scenarios involving varying degrees of

population bottleneck effects, thereby enhancing the relevance of our findings to species experiencing population decline. Noteworthy is our decision to relocate certain assertions—those not immediately deducible from the illustrative figures—to the "Discussion" section for a more in-depth exploration.

These modifications are a testament to our diligent efforts, and we hope they effectively address and alleviate your concerns.

Referee #2

Comments to the Author

Zhou et al. introduce in this manuscript an interesting new set of statistics to estimate mutational load based in linkage disequilibrium among deleterious loci. The theoretical background is solid, the Methods section is well explained and the mathematical approach rather robust, simulations are correctly run and the examples are (in principle) correctly chosen. However, I have some concerns about the structure of the paper, especially after the section "The new LD statistics have great consistency...". Some claims sparkled in the Results section do not have enough evidence to be sustained with the present analyses or look like they would fit better in Discussion. Some implications of the results of the paper cannot be drawn with the evidence presented. Also, most figures and supplementary figures have footers which are not well worded and would need to be expressed more clearly. All in all, I think that this manuscript brings a couple new set of statistics that would be of great use for the scientific community that works in related fields; however, it needs a deep revision of some sections and some re-thinking of the theoretical framework. Comments below.

Response:

Thank you very much for remarking our theoretical background and simulations. We recognize the valuable contribution of LD statistics in assessing mutational load and are committed to enhancing the application of LD statistics for demographic analysis and animal conservation efforts. Indeed, we were pleased to find that the variation in LD across different evolutionary factors—such as recombination rate, selection coefficient, synergistic epistasis, population bottlenecks, and population size—aligned closely with our expectations.

We appreciate your pointing out instances where certain claims in our manuscript may not have been fully substantiated by our results. In response, we have meticulously reviewed the manuscript and relocated statements like “The LD pattern of recessive deleterious mutations is due to the complex interplay of heterosis effect ...” and “... Thus, the LD value of a single pair of loci can be confusing. Therefore, we suggest using the relative value of the mean LD instead,” to the Discussion section. In the revised version, Figure 2 systematically delineates the influence of various factors on LD, as demonstrated through our simulations. We have carefully ensured that only conclusions directly supported by Figure 2 remain in the

“Results” section, while insights derived from these findings are elaborated upon in the “Discussion” section, specifically under the subsection “The Feasible Effect of Several Factors on LD” (lines 519-552).

Furthermore, we have undertaken a thorough reorganization of the figures' structure and color scheme in the revised version, aiming to enhance clarity and readability. Notably, the color gradients in Figure 2 and similar figures have been adjusted to feature contrasting colors for better differentiation, and the figure captions have been comprehensively annotated.

Below, you will find a detailed point-by-point response. We earnestly hope that our amendments align with your feedback and successfully address your concerns.

Comment 2-1:

181 - "and was implemented into SliM v3.0" this is redundant to the previous mention of SliM and does not need to be here.

Response:

We agreed and removed the statement “and was implemented into SliM v3.0” here.

Comment 2-2:

183 - "For each simulation, we sampled 1,000 individuals (100 individuals when only 100 individuals were set in the simulation) from the African population in the human demography." This means that for the simulation with 10,000 individuals, 10% of them are sampled as subset. This can greatly increase the speed of calculation of statistics, but there are an increase of biases and false positives due to Wahlund effect in some loci. In my personal experience, it is better to test if subsetting the population to 10% is going to affect the estimation of statistics before proceeding. I would recommend, for the sake of time, to perform such test with the population of 1,000 by sampling 100 individuals and seeing if the estimations are consistent to what you found by sampling 1,000 out of 1,000.

Response:

Thank you for bringing this to our attention. In the previous iteration of our study, we did consider the potential impact of resampling and posited that it would not significantly influence the outcomes for two main reasons. First, while the Wahlund effect is generally non-negligible across sampled populations for any specific locus or pair of loci, our application of LD statistics does not focus on individual SNPs. Instead, it targets a curated group of SNPs, such as a set of loss-of-function (LoF) loci within a population. This approach inherently mitigates potential biases from this effect. Second, to further safeguard against bias, for each parameter tested, we conducted approximately 100 simulation runs and used the mean value as the representative outcome, enhancing the reliability of our findings.

We acknowledge the need to scrutinize how subsetting the population to 10% influences the assessment of population size effects on LDcor. The implications of sampling 100 out of 1,000 versus 1,000 out of 1,000 may indeed differ from those of

sampling 1,000 out of 10,000 versus 10,000 out of 10,000. To efficiently estimate the impact of resampling while also considering time constraints, we introduced three different resampling scales (10%, 7%, and 4%) in our revision, based on simulations using model 1 with a base population of 10,000 individuals (illustrated in Fig. R2). Our findings indicate that the general trend persists—LD typically decreases with increasing negative selection intensity. Interestingly, we observed that the LD value consistently increased following the subsampling process for any given selection coefficient. This observation has been incorporated into Fig. 2B (represented by gray lines) in the revised version.

Fig R2: The effect of sub-setting the population size from 10000 to 1000, 70, or 400. At the end of simulation (model 1), we sampled subsets with different size (e.g., 10000, 1000, 700, and 400, represented by different colors) for each run. Mean value and standard error mean of LDcor under these sampling procedures were compared.

Comment 2-3:

224 – “For each population, SNPs with DP in the upper 5% (giant panda:426, rhesus macaque:302, golden snub-nosed monkey: 786, tiger: 531, and sheep: 412) or lower 5% (giant panda: 200, rhesus macaque:4, golden snub-nosed monkey: 181, tiger: 197, and sheep: 204)” – can the authors provide a reason why they have followed this criterion? Normally the DP filter is set up at a minimum of 3-5X per sample and 200% (or 250%) of the average depth of coverage.

Response:

Thank you for your suggestion. In response to the varied coverage across the empirical datasets collected from multiple sources, we initially applied a percentage-based criterion to exclude SNPs within the upper or lower 5% of depth. Following your recommendation, we have revised our approach to implement DP (Depth of Coverage) filtering criteria, setting a minimum threshold of 3X per sample and a maximum of 250% above the average depth of coverage in the updated version.

This adjustment has led to changes in the variant numbers with different functional consequences. Correspondingly, we have updated Fig. 4 to reflect these modifications accurately.

NEW

Species	Type	Function	Number
Tiger	synonymous	synonymous_variant	17064
	missense	missense_variant	14419
	LoF	stop gained	125
	LoF	stop lost	17
	LoF	splice acceptor variant	59
	LoF	splice donor variant	108

Table EV3: Variant numbers of different functional consequences (tiger).

Fig. 4: LD statistics for a select set of mammal populations (tiger).

PREVIOUS

Species	Type	Function	Number
Tiger	synonymous	synonymous_variant	22893
	missense	missense_variant	18743
	LoF	stop gained	189
	LoF	stop lost	21
	LoF	splice acceptor variant	76
	LoF	splice donor variant	128

Table EV2: Variant numbers of different functional consequences (tiger).

Fig. 4: LD statistics for a select set of mammal populations (tiger).

Comment 2-4:

285-289 *"The LD pattern of recessive deleterious mutations is due to the complex interplay of heterosis effect (associative overdominance) whereby these mutations are masked from selection in the heterozygous state. This effect may be exaggerated if there is a small degree of recombination and the deleterious alleles arise on the same haplotype, leading to positive L_{dcor} ." This sounds like a precipitate conclusion and the connection to the results is not clear. I would appreciate if authors could elaborate more about why this is the case and cite a similar case from the reference 35, stating what is the connection between this conclusion and the reference. Also, this would fit better in Discussion.*

Response:

Apologies for any confusion caused earlier. Recessive alleles do not affect an organism's fitness when present in a heterozygous state because the effect of the recessive allele is overshadowed by the dominant allele. Consequently, deleterious recessive alleles tend to accumulate since they are shielded from negative selection's full impact, leading to a greater likelihood of remaining in a population compared to deleterious dominant alleles. This scenario introduces more variability in heterozygosity (referred to as the heterosis effect) and linkage disequilibrium (LD). Additionally, this heterosis effect may be amplified in the context of a longstanding low recombination rate, which impedes efficient crossover. In summary, recessive alleles are more likely to persist in heterozygosity and more prone to positive LD. The degree of LD among a group of recessive alleles can vary significantly.

Our analysis that associative overdominance, resulting from harmful recessive mutations, enhances the marginal fitness of marker heterozygotes and mitigates the selection pressure on LD aligns with the findings of Pamilo, P., & Pálsson, S. (1998) in their study "Associative overdominance, heterozygosity and fitness". Their research, which involved extensive simulations, explored how the heterosis effect (associative overdominance) influences heterozygosity, fitness, and genetic diversity. They noted that associative overdominance induced by harmful recessive mutations plays a role in preserving genetic variation. However, this effect diminishes in large populations and with an increase in the rate of crossing over.

Besides, the term "The LD pattern of recessive deleterious mutations....., leading to positive L_{dcor} ." has been moved to "Discussion" (lines 522-527).

Comment 2-5:

296 - 301 *"For any selection coefficient, L_{dcor} always decreases monotonically as the synergistic epistasis level increases. Under the same synergistic epistasis level, L_{dcor} usually decreases monotonically as the negative selection coefficient increases. Both negative selection and synergistic epistasis can increase negative LD, suggesting that an individual locus may not be the appropriate unit of selection"*

Figure 2 is not appropriately addressed in this section (until line 321). If such a conclusion "Both negative selection and..." is to be drawn, it should be at the end of the entire section, or better subdivide Fig. 2 into 3 sections (A, B, C) and correctly point out to each sub-figure with each logical conclusion and explanation of results per separate. The text is not very solid, since after this last statement we have "We next examined... (line 302)". I would suggest redoing this section, addressing the subfigures of Figure 2 one by one.

Response:

Thank you for your insightful suggestion. In response to your feedback, we updated a new version of figure 2, with a distinct focus on four crucial factors: the levels of epistasis (now Figure 2A, previously the left part of Fig. 2A), the influence of population size (now Figure 2C, previously the right part of Fig. 2A), the significance of loci distances (now Figure 2D, previously Fig. 2B), and the implications of a population bottleneck (introduced as Figure 2B). We delve into the observed patterns associated with various selection coefficients for each of these factors. Consequently, we have devoted specific sections within the "Results" (lines 298-378) to thoroughly elucidate the impacts of these factors on LD, based on the models tested.

NEW

Fig. 2: LDcor with different levels of synergistic epistasis, bottleneck, demography, and maximum derived allele frequency. (A) We simulated different synergistic epistasis levels (values in 0, 10^{-3} , 10^{-2}) as well as different negative selective coefficients (0, 10^{-1} , 10^{-2} , 10^{-3} , and 10^{-4}) for each model (10,000 individuals) with a recombination rate of 1×10^{-8} crossovers per base pair per generation. (B) In order to

compared the LDcor of varying magnitudes of bottleneck (based on model 1), we simulated different scaled population size (from 10000 to 1000: bottleneck 1, 700: bottleneck2, and 400: bottleneck 3) as well as different duration time (0, 2, 6, and 10 generations). And the dark gray line is the results of no-resampling (10000) at time 0 (e.g., time 00). **(C)** The effect of different models and demography (sample1(10000,1000): a simulation with 10000 individuals and a subset of 1000 individuals in calculating; sample2(1000,1000): a simulation with 1000 individuals and a subset of 1000 individuals in calculating; sample3(100,100): a simulation with 100 individuals and a subset of 100 individuals in calculating) on LDcor for different selection coefficients. **(D)** For all three models, we examined the LDcor of variants with selection coefficients of 0 and beyond a certain distance away from each other (10bp, 100 bp, and 1000 bp). The LDcor values were normalized by reducing the LDcor of variants without distance limitation.

PREVIOUS

Comment 2-6:

303 "In simulations with 10,000 individuals evolving" - delete evolving.

Response:

Done

Comment 2-7:

305 - 308: "For any specific selection coefficient, model 1 had the largest Ldcor value, followed by model 3 and model 2, suggesting that population expansion may produce less LD while increased gene flow contributes to increases in LD". This is not what

can be seen in Fig. 2A (left). Model 3, for $s = -10^{-4}$, it looks like it has a higher value than model 1. If this is not the case, since the graph is not entirely clear I would suggest authors to mention the numerical values and explain how higher they are to each other.

Response:

Apologies for any confusion created earlier. The clarification pertains to the initial segment of Figure 2C, not Figure 2A, as mistakenly mentioned. Figure 2C is split into three subfigures, each aiming to represent the observed LDcor patterns across varying population sizes, specifically within three distinct models (depicted using different colors for clarity: model 1 in blue, model 3 in purple, and model 2 in red). In the first subgraph of Fig. 2C, it is generally observed that the blue line (model 1) appears above both the purple (model 3) and red (model 2) lines.

It seems that our initial arrangement of Figure 2 might have led to some misunderstanding. To enhance clarity in the updated version, we have segregated these effects, showcasing them individually in Fig. 2A (which now corresponds to what was previously the left side of Fig. 2A) and Fig. 2C (formerly the right side of Fig. 2A). This reorganization and the related comparison are further elaborated in Comment 2-5.

Fig. 2C The effect of different models and demography (sample1(10000,1000): a simulation with 10000 individuals and a subset of 1000 individuals in calculating; sample2(1000,1000): a simulation with 1000 individuals and a subset of 1000 individuals in calculating; sample3(100,100): a simulation with 100 individuals and a subset of 100 individuals in calculating) on LDcor for different selection coefficients.

Comment 2-8:

308: *"Genetic drift diminished greatly after population expansion..." This is the first reference to genetic drift in the entire manuscript and there is no way of the authors to detect it. This argument is not solidly presented.*

Response:

Thank you for pointing out this deficiency. We have removed this statement.

Comment 2-9:

309-310: *"In model 3 both gene flow and differences in allele frequency among subpopulations contributed to LD" Although I understand the meaning of this sentence and why the authors mention it, this feels like it does not fit in this paragraph. This is not a directly viewable result and I would suggest moving it to Discussion.*

Response:

Thank you for this suggestion. We have revised it to **"The observed higher LD values in model 3 compared to model 2 are likely attributable to gene flow and variations in allele frequency across subpopulations, which are known factors contributing to LD⁶¹⁻⁶⁴ (Figs. 2B, EV4B)."**, and moved it to **"Discussion"** in the updated version (lines 535-537).

Comment 2-10:

335 - *"While, LD" - Perhaps you mean "Meanwhile, LD..."?*

Response:

Changed as suggested.

Comment 2-11:

336 - *"when loci far located" - I would add "to each other". Also, it looks like Figures S6 and S7 and all text associated to them should be moved to section "Estimation of LD in six mammalian species".*

Response:

We changed the statement to **"when loci far located to each other"** in the updated version. Figures EV6 and EV7, which were previously referred to as Figures S7 and S8, are indeed focused on examining the influence of loci distances on LD values, rather than presenting data on six mammals as initially mentioned. Apologies for the confusion.

Comment 2-12:

355 - 364. *This paragraph should be in Discussion.*

Response:

Changed as suggested.

Comment 2-13:

376 "*while Ldcor was always decreased*" it should be "*while Ldcor always decreased*"

Response:

Changed as suggested.

Comment 2-14:

389 - 391: "*Three of these are recognized by the IUCN as threatened species: the goldenSM, tiger, and giant panda.*" Aesthetically, it would be better to include this information within the previous sentence or in *Methods*, not here.

Response:

We added a new threatened species gorilla and have moved the statement to "Materials and methods" in new version (lines 233-234).

Comment 2-15:

394 - 400 "*In our results...*" There is no reference to any figure or table. Where are these results extracted from?

Response:

The sentence "In our results..." here should be related to Fig. 4, and we added this reference in the updated version.

Comment 2-16:

401 - 403 "*For most of the nondomesticated populations, the distribution of rare (e.g., derive allele count ≤ 5) LoF allele sets were underdispersed relative to synonymous and missense sets of the same derived allele count*" I don't see clear how this conclusion is extracted from Table 2 and Fig. 4.

Response:

Sorry for this confusion. We extract the conclusion that "For most of the non-domesticated populations, the distribution of rare (e.g., derive allele count ≤ 5) LoF allele sets were under-dispersed relative to synonymous and missense sets of the same derived allele count" directly from the widely exited shadings in Fig. 4 (gray shadings) and putatively smaller LDcor values in LoF sets in Table 2 (red numbers).

In Fig. 4, the LD values of three groups of loci sets (e.g., Lof, missense, and synonymous) fluctuate across different maximum derived allele counts (MDAC, shown on the x axis), with the under-dispersion in LD of LoF loci set (relative to the LD value of synonymous loci set) highlighted by gray shading. In Table 2, smaller LDcor values in LoF are observed in more than half non-domesticated populations (tiger, goldenSM, human (CEU), and gorilla).

Fig. 4: LD statistics for a select set of mammal populations.

Table 2. Linkage disequilibrium among different allele sets of importance in the tiger, rhesus macaque, golden snub-nosed monkey, giant panda, human (CEU), and sheep genomes. Here, only alleles with an allele count no more than 5 are considered (MDAC=5).

Species	Type	Nloci	Mean	LDcor	LDcorabs	NetLD ²³
Tiger	LoF	61	5.75	3.56E-02	1.74E-01	2.83E-03
	missense	2890	278.25	3.84E-02	1.72E-01	3.55E-03
	synonymous	3138	312.16	4.46E-02	1.78E-01	4.39E-03

Macaque	LoF	89	3.47	4.66E-03	6.99E-02	1.68E-04
	missense	3687	135.00	4.22E-04	6.25E-02	1.43E-05
	synonymous	3312	124.58	6.38E-04	6.42E-02	2.21E-05
GoldenSM	LoF	109	10.21	-3.14E-03	1.40E-01	-3.94E-04
	missense	4748	451.15	1.86E-03	1.45E-01	2.10E-04
	synonymous	4349	421.03	1.49E-03	1.45E-01	1.68E-04
Panda	LoF	14	1.04	1.64E-02	1.30E-01	1.27E-03
	missense	551	37.84	5.64E-03	1.14E-01	3.62E-04
	synonymous	405	28.02	8.96E-03	1.18E-01	5.30E-04
CEU	LoF	66	1.92	-1.41E-03	4.99E-02	-4.37E-05
	missense	2816	83.26	1.09E-04	5.28E-02	3.97E-06
	synonymous	2034	62.71	3.54E-05	5.50E-02	1.24E-06
Sheep	LoF	759	90.48	2.96E-03	1.62E-01	3.80E-04
	missense	18239	2157.37	7.58E-04	1.62E-01	9.15E-05
	synonymous	18796	2240.89	7.15E-04	1.61E-01	9.06E-05
Gorilla	LoF	62	6.00	-1.43E-03	1.36E-01	-3.18E-05
	missense	2793	280.16	3.02E-03	1.47E-01	7.56E-04
	synonymous	2647	269.26	2.58E-03	1.48E-01	6.29E-04

Comment 2-17:

404 - "This pattern was consistent even when allele frequency was controlled (Fig. S8)" Perhaps you mean Fig. S6?

Response:

The correct reference figure here should be Figure EV6 instead of Figure EV8 in the previous version. In the update version, we have renamed Fig. S6 as Figure EV8 in the SI (Figs. EV6, EV7, EV8 are previously Figs. S7, S8, S6), and the statement regarding Figure EV8 is now accurate.

Comment 2-18:

410 - 411 "similar situation was observed in human, goldenSM (Endangered), and the giant panda (Vulnerable)." However, LoF Ldcor values are higher than synonymous and missense in goldenSM above DAF > 5. I would suggest adding the DAF scale to each of the subgraphs of Ldcor and Ldcorabs, since it is not easy to see.

Response:

Thank you for this suggestion. We agree and keep the DAF scale for each of the subgraphs. Besides, we take gray shading for the case that LoF more under-dispersion (lower) than synonymous. The area without gray shading thus indicates case that LoF more over-dispersion (higher) than synonymous. Fig. 4 has been presented in Comment 2-16.

Comment 2-19:

The whole paragraph (405 - 420) should address properly and clearly between which values of DAF these trends of LDcor are higher or lower for LoF.

Response:

Thank you for this suggestion. We agree that it is important to specify the range of DAF where trends of LDcor are higher or lower for LoF. The regions shaded in gray in Fig. 4 represent cases where LDcor values are lower for LoF, while the unshaded areas represent cases where LDcor values are higher for LoF. We have provided this information for each population separately (lines 416-438):

“We subsequently examined LD metrics within Loss-of-Function (LoF) loci sets of these mammals. Within the sheep population, LDcor values exhibited significant over-dispersion in LoF and missense loci sets when compared to synonymous datasets (**Table 2, Fig. 4**). Conversely, for most non-domesticated species, rare LoF allele sets (e.g., MDAC ≤ 10) showed under-dispersion relative to synonymous and missense sets sharing the same MDAC metrics (**Table 2, Fig. 4**). In the human (CEU) populations, both LDcor and LDcorabs scores were consistently the lowest within LoF variants at appropriate MDAC levels, indicating that LoF loci experienced the least linkage load and were most strongly influenced by negative selection. This effect is evident not only in the selection coefficient but also in the combined impact of synergistic epistasis and other selection-related forces across various variant categories. This trend was similarly observed in several other species: gorillas (Critically Endangered), rhesus macaques (Least Concern), golden snub-nosed monkeys (Endangered), and giant pandas (Vulnerable). In gorillas, LD scores were generally the lowest for LoF variants, with the exception at MDAC=4. For rhesus macaques, LDcor and LDcorabs scores were typically highest in LoF variants. In golden snub-nosed monkeys, both statistics were notably lowest in LoF when MDAC ≤ 6 , but increased at other ranges. Giant panda populations saw the lowest scores in LoF at MDAC of 2, 3, 6, or 8, but showed higher scores in LoF for MDAC >4. In tiger populations, both LDcor and LDcorabs were generally lower in LoF compared to synonymous variants (MDAC>3), highlighting a varied impact of negative selection across species and suggesting nuanced evolutionary pressures at play. These patterns were consistent even when allele frequency was controlled (**Fig. EV8**).”.

Comment 2-20:

429 "suggesting that giant panda" either "suggesting that giant pandas" or "suggesting that the giant panda".

Response:

We change this statement as “suggesting that the giant panda” in our revision.

Comment 2-21:

432 - 435 "We further scaled the population size of giant pandas to 10-fold smaller

and then evolved 100 generations. The results indicate that the LD of slightly deleterious variants was mixed with that of neutral variants (Fig. 5), suggesting an increase of potential genetic risk." This paragraph is terribly worded and obscure. A different way to write it would be: "We further scaled the population size of giant pandas down 10-fold in our simulation and ran it for 100 generations. The results indicate that the LD values/trend of slightly deleterious variants was indistinguishable from that of neutral variants". The explanation of increase of genetic risk is not appropriately done.

Response:

Thank you for your advice. We have incorporated your improved wording: "We further scaled down the population size of giant pandas by 10-fold in our simulation... indistinguishable from that of neutral variants." We inferred an increase in genetic risk because deleterious variants showed no discernible negative selection pressure.

NEW

Fig. 5: Simulation results of LD with different selection coefficient and max derived allele count in giant panda and golden snub-nosed monkey.

PREVIOUS

Fig. 5: Simulation results of LD with different selection coefficient and max derived allele count in giant panda and golden snub-nosed monkey.

Comment 2-22:

441 “ignores the linkage among different loci” better “ignored the importance of linkage”

Response:

Revised as suggested.

Comment 2-23:

444 “ignores the direction”. Of what? Perhaps of linkage disequilibrium?

Response:

This sentence should be “ignores the direction of linkage disequilibrium”.

Comment 2-24:

452 “ V/V_A ” This is the first mention to this equation in the entire paper.

Response:

“ V/V_A ” is accompanied by reference at line 869 and the formula statement following NetLD in the “Materials and method” (lines 129-131):

“Then, V/V_A was calculated as $(n(n-1)/2 * \text{NetLD} + V_A)/V_A$, where V_A represents additive variance of all variants in the tested set and V represents variance for the sum of all variants²³”

Comment 2-25:

490 - 496 “We tested... are small (Fig. S10)” This entire section belongs to Results.

Response:

We moved this section “We tested... ... are small (Fig. EV10)” to “The standardization of LDcor and LDcorabs” of “Results” in the update version (lines 457-464).

Comment 2-26:

499 - 500 "explain how humans have accommodated high mutation rates over our evolutionary history." I would need a reference for this ascertainment.

Response:

We have added reference (“Negative selection in humans and fruit flies involves synergistic epistasis”) here in our new manuscript.

Comment 2-27:

528 “Both the two” is redundant.

Response:

This has been removed.

Comment 2-28:

I would change "table" for "Table" in all references in text.

Response:

Changed as suggested.

Comment 2-29:

Fig. 2 "(A) We simulated different synergistic epistasis levels (0,10-1,10-2, 10-3, and 10-4) as well as different negative selective coefficients (0, 10-2, 10-3, and 10-4) for each model (10,000 individuals) with a recombination rate of 1×10^{-8} crossovers per base pair per generation (left). The effect of different models and demography (Sample 1 (10,000,1,000): with 10,000 individuals evolving and 1,000 individuals used in the calculation; Sample 2 (1,000,1,000): with 1,000 individuals evolving and 1,000 individuals used in the calculation; Sample 3 (100,100): with 100 individuals evolving and 100 individuals used in the calculation) on Ldcor for different selection coefficients (right)" This figure is not appropriately explained. Both the organization of the image and the footer are confusing, since (A) left is depicting something entirely different to (A) right. I would subdivide the figure in 3 (A, B, C). Also, there is no reference to what each line means, neither at the footer nor in the image.

Response:

Thank you for your suggestion. In the updated version, Figure 2 has been reorganized for enhanced clarity. Furthermore, we have introduced a legend in the figure's footer to elucidate the meaning of the various colors used. This revised Figure 2 is detailed in Comment 2-5.

Comment 2-30:

Fig. 3 "The square root of classical r^2 (signed by D)" - you should find a better way to

express this statistic instead of a piece of text. Perhaps "signed". The colors of this figure are difficult to read, we would need them to be more intense. Also, I would suggest using a colorblind-friendly palette. Also, I would eliminate the explanatory text "Ldcor and Ldcorabs show great...". This would fit in the main text, not here.

Response:

Thank you for your suggestion. We have changed the expression "The square root of classical r^2 (signed by D)" into refined language "(signed $\sqrt{r^2}$)" in Fig. 3 as your advice. We also changed the colors in pictures. And, the text "Ldcor and Ldcorabs show great..." is the name of the figure.

Fig. 3: LDcor and LDcorabs show great consistency with classical LD statistics.

Comment 2-31:

Fig. 4 "codingsynononly" is a mistake. It should be "synonymous"

Response:

Revised as suggested.

Comment 2-32:

Fig. 5 "Careful attention must be paid to the population genomic health of golden snub-nosed monkey (left)." What is "population genomic health"?

Response:

Sorry for this deficiency. We were thinking to use "genomic health" to describe genomic sustainability, the opposite of any risk in individual survival or reproduction caused by genetic factor. During revision, we removed this word and revised it as (lines 56-58):

"This has led researchers to question the sensitivity of these estimators in gauging genetic burden and to reconsider the presumed correlations between population size and genetic diversity."

Comment 2-33:

Fig. S2 needs more resolution. It is hard to see differences between the dark dotted line with dots and the line with "X". I would also change the name of "0" simulations and "-0.01" simulations. Perhaps naming the datasets would help mention them more easily.

Response:

We concur with your suggestion to enhance clarity by labeling the datasets depicted in Figure EV2 with specific names. In the updated version, the blue solid line representing LD from simulations with no change is now labeled "S"; the purple dotted line, indicating simulations with a -0.01 effect, is labeled "D"; and the red dotted line marked with "X", which represents simulations at 0 but with the allele frequency spectrum matching that of the -0.01 simulations, is labeled "SR."

Figure EV2: Mean LDcor and LDcorabs values with different selection coefficient when allele

frequency spectrum is controlled. (A) We resampled variants from the 0 simulations with the same allele frequency spectrum of -0.01 simulations and compared LDcor. The blue solid line with represents LD of 0 simulations (named as S), the purple dotted line with dot represents -0.01 simulations (named as D), and the red dotted line with "×" represents 0 simulations but with the same allele frequency spectrum of -0.01 simulations (named as SR). The LDcor of SR is always between that of S and D. (B) The LDcorabs values in S, D, and SR under varying max derived allele count.

Comment 2-34:

Fig, S3 "(A) For all three models (model1: constant population size; model 3: with population expansion and gene flow; model 2: with population expansion), we did simulations 120 runs for each recombination rate and each selection coefficient. The point and the error bar represent mean value of Ldcor and a standard error mean. (B) For all three models, we did simulations 120 runs for each recombination rate and each selection coefficient. The point and the error bar represent mean value of Ldcorabs and a standard error mean."

These are not appropriate explanations for this figure. A better way of expressing what is in the figure would be: "(A) Ldcor values in three models (model1: ; model3: ...). We ran 120 simulations for each recombination rate and each selection coefficient. The point and the error bar..."

Response:

We have incorporated your improved wording in new version: "(A) Ldcor values in three models (model1: ; model3: ...). We ran 120 simulations for each recombination rate and each selection coefficient. The point and the error bar..."

Comment 2-35:

Fig. S4 – same as with Fig. 2, it should be divided in 3 sections and its results should be more precisely addressed in main text.

Response:

We rearrange the organization of **Figure EV4** and add footer for different colors in the updated version. Also, the results of subgraphs in Fig. 2 have been shown separately.

NEW

PREVIOUS

Comment 2-36:

Fig. S5 "with 10000 individuals evolving and 1000 individuals in calculating" This is not a good way to express results. I would say, "a simulation with 10000 individuals and a subset of 1000 individuals".

Response:

Revised as suggested.

Comment 2-37:

Fig. S8 - A notation of only 2 decimals would be better in the third set of graphs (right).

Response:

We have updated the notation in Figure EV7 (formerly Fig. S8) to display values with only two decimal places for greater clarity. The revised Figure EV7 is detailed in Comment 2-11. Thank you for your advice.

Referee #3:

The authors introduce new statistics to detect and compare global linkage disequilibrium of deleterious mutations across species, using unphased genotypes. The study employs simulation analyses to validate the effectiveness of the new statistics in standardizing differences among mutations with varying allele frequencies. The analysis focuses on six mammal species, namely sheep, tigers, golden snub-nosed monkeys, giant pandas, humans, and rhesus macaques. The results reveal overdispersion of deleterious mutations in sheep, tigers, and golden snub-nosed monkeys, suggesting less effective negative selection. Conversely, strong negative linkage disequilibrium of deleterious mutations is observed in giant pandas, humans, and rhesus macaques. The introduction of new statistics to assess global linkage disequilibrium is a notable contribution, providing a novel approach for evaluating selection intensity. However, some aspects should be clarified for a more comprehensive understanding.

Response:

We are truly thankful for the summary of our study and for highlighting our contributions, especially our novel approach for assessing selection intensity. In response to your suggestions, we have thoroughly reviewed and incorporated additional details in the updated version. Below you will find a detailed point-by-point response. We hope that we have accurately understood your queries and fully addressed your concerns.

Comment 3-1:

1) The rationale for choosing the six specific mammal species and the implications of their observed selection patterns should be elaborated. Why the selection of the species studied (all unrelated)? It would be interesting to see also a comparison between some closely related species using shared (ancestral) loci/genes, such as human and other close ape species, for example. It could give more intuitive insights combined with clear known evolutionary path.

Response:

Thank you for highlighting this point. We selected four wild populations, each representing a distinct conservation status: the rhesus macaque, categorized as LEAST CONCERN and the most widely distributed primate after humans; the giant panda, currently classified as VULNERABLE after previously being listed as ENDANGERED, and native to China; the golden snub-nosed monkey, now considered ENDANGERED after previously being VULNERABLE, also endemic to China; and the tiger, classified as ENDANGERED, representing the largest extant cat species. Notably, the giant panda and snub-nosed monkeys serve as flagship species within the field of conservation biology. Additionally, our laboratory and department boast a longstanding history of research on these species, granting us comprehensive

insight into their statuses and backgrounds. Beyond these wild populations, our study also encompassed the domesticated sheep population and the extensively examined human population for comparative analysis.

Heeding your advice, we have expanded our comparison to include the gorilla, a **CRITICALLY ENDANGERED** species closely related to humans. Our analysis revealed that the LDcor values for LoF mutations in gorillas consistently exceed those in humans (<0), likely attributable to the gorilla's ongoing population decline and habitat reduction. These findings and related discussions have been incorporated into the updated manuscript (lines 429-431, 579-589).

Comment 3-2:

2) The normalization of LD based on synonymous mutations (l. 157-159, l. 472), would require some further elaboration. Could this be introducing some bias by assuming that synonymous mutations are always neutral and homogeneous. For instance, currently selection tests already assume the variability of synonymous mutation rates. Would it be also relevant here to take it in consideration?

Response:

This raises a valuable point for discussion. Currently, selecting a neutral reference for empirical analyses presents challenges, including the potential introduction of bias. The question of whether synonymous mutations are genuinely neutral has been a hot topic of recent debate. Shen et al. (2022) posited that in a study of representative yeast genes, most synonymous mutations were strongly non-neutral, igniting widespread interest in the neutrality of these mutations among researchers^{68,69}. However, the generalizability of Shen et al.'s findings to other organisms remains to be confirmed. In a subsequent study, Kruglyak et al. (2023) argued that evidence supporting the neutrality of synonymous mutations is not robust, pointing to technical limitations in experimental design and replication⁷⁰.

The investigation into the neutrality of synonymous mutations is ongoing. Distinguishing synonymous from non-synonymous variants may soon require the same level of experimental evidence, though such data are currently sparse^{71,72}. Synonymous variants differ from non-synonymous ones, notably in their distribution of allele frequencies. They are subject to significantly less constraint, making them more likely to be neutral⁷³.

Given the scarce availability of experimentally verified silent mutations and their generally weaker selection pressures, we have chosen to use synonymous mutations as a neutral or nearly neutral reference point for our empirical analysis of global linkage disequilibrium (LD) indexes (lines 613-630). Nevertheless, in response to your concerns, we have acknowledged this limitation in our discussion section, stating (lines 568-574): “Furthermore, while the standardization of LDcor and LDcorabs (to nLDcor and nLDcorabs, respectively) was implemented to mitigate the impacts of hitchhiking, genetic drift, demographic factors, or data quality, the normalization

process inherently presumes synonymy in mutations — assuming them to be neutral and uniform across instances. This assumption has increasingly come under scrutiny. As such, the development of future LD estimators should aim to differentiate or measure the variability inherent in synonymous mutations.”

Comment 3-3:

3) In the methods "ancestral genome reconstruction" (l. 235) it is mentioned closely related species, but no further detail provided (what species, divergence times, etc...).

Response:

Thank you for your suggestions. We have added detailed information of this part, including closely related species, divergence times, reference genomes, and exact pipeline, in the “Materials and methods” as well as Table EV2 of our new version (lines 261-271):

“We established the ancestral allele by leveraging the genome sequences of targeted species and their closely related counterparts (**Table EV2**). For each species, a one-to-one genome alignment with its closely related species was produced by LAST v2.31.1^{44,45}, following the parameters and commands used in generating “2017 human-ape alignments”. Subsequently, for biallelic SNPs that were fully represented in the VCF file (i.e., without missing sites), we extracted precise coordinate conversion details using MafFilter v1.3.1⁴⁶. When the reference allele of the related reference species corresponded to either the reference or alternative allele of the target species, it was designated as the ancestral allele. For the CEU human population and additional gorilla samples, we utilized chain files from UCSC and applied LiftOver⁴⁷ or direct coordinate conversion information retrieval.”

Table EV2: The information of ancestral reconstruction for six mammals.

Species	reference genome	close species	reference genome	divergence time
Homo sapiens	hs37d5	Gorilla gorilla	gorGor3	8.60 MYA
Ailuropoda melanoleuca	ailMel1	Canis lupus familiaris	CanFam3.1	45.1 MYA
Ovis aries	Oar_v3.1	Bos taurus	ARS_UCD1	21.6 MYA
Rhinopithecus roxellana	Rrox_v1	Macaca mulatta	Mmul_8.0.1	17.75 MYA
Panthera tigris	panTig1.0	Canis lupus familiaris	CanFam3.1	55.4 MYA
Macaca mulatta	Mmul_8.0.1	Homo sapiens	hg19	28.82 MYA
Gorilla gorilla	gorGor3	Homo sapiens	hg19	8.60 MYA
Canis lupus familiaris	CanFam3.1	Ailuropoda melanoleuca	ailMel1	45.1 MYA

Comment 3-4:

4) The term "narrowing purified selection" might benefit from further clarification to enhance reader comprehension.

Response:

We have changed this description to “relaxed negative selection” in our new manuscript.

Comment 3-5:

5) Consider introducing the two new statistical approaches, namely the averaged pairwise standardized covariance (LDcor) and the averaged absolute pairwise standardized covariance (LDcorabs), in the abstract for a more comprehensive overview.

Response:

In the updated version of our abstract (lines 26-30), we have included an introduction to LDcor and LDcorabs to provide a clearer announcement of these concepts:

“Here, we introduced new statistics, LDcor and LDcorabs, which allows us to detect and compared global linkage disequilibrium of deleterious mutations across species using unphased genotypes. These statistics measure averaged pairwise standardized covariance and absolute covariance, respectively, and help standardize mutation differences based on allele frequencies to reflect selection intensity.”.

Comment 3-6:

6) Maintain consistency by either using "purifying" or "negative" selection throughout the manuscript.

Response:

Thank you for your advice. We used “negative” selection throughout our new manuscript.

Comment 3-7:

7) Before comparing with your statistical methods, introduce both classical r^2 and D' (signed by D) and NetLD.

Response:

We introduced both classical r^2 , D' , and NetLD in “Materials and methods” in our revision (lines 113-133).

Comment 3-8:

8) To facilitate readers, especially those interested in employing your statistical methods, consider providing the R script mentioned in the manuscript. This would enhance transparency and reproducibility.

Response:

We have uploaded the related scripts to github (<https://github.com/Chunyan-Hu/LDcor.git>).

Comment 3-9:

9) Overall the English needs improvements. There are many small mistakes and grammatical errors throughout the manuscript that would need revision.

Response:

Thank you for your suggestions. We have sent the manuscript to native speaker to polish our language and we hope there is no obviously grammatical errors now.

Dear Dr. Zhou

Thank you for the submission of your manuscript to EMBO Reports. We have now received the full set of referee reports as well as referee cross-comments that are all pasted below. I would like to once again apologize for the atypically long time it took us to review your manuscript.

As you will see, the referees acknowledge that the findings are potentially interesting. However, Referee #1 has strong reservations and concerns that require thorough and significant revision from your side.

I would thus like to invite you to revise your manuscript with the understanding that the referee concerns must be fully addressed and their suggestions taken on board. Please address all referee concerns in a complete point-by-point response. Acceptance of the manuscript will depend on a positive outcome of a second round of review (either by external experts or by myself alone). It is EMBO reports policy to allow a single round of major revision only and acceptance or rejection of the manuscript will therefore depend on the completeness of your responses included in the next, final version of the manuscript. In your case we decided to make an exception and allow a second round of revision, but please be advised this could not continue to a 3rd round.

We realize that it is difficult to revise to a specific deadline. In the interest of protecting the conceptual advance provided by the work, we recommend a revision within 3 months (16th Oct 2024). Please discuss the revision progress ahead of this time with the editor if you require more time to complete the revisions.

- 1) A data availability section providing access to data deposited in public databases is missing. If you have not deposited any data, please add a sentence to the data availability section that explains that.
- 2) Your manuscript contains statistics and error bars based on $n=2$. Please use scatter blots in these cases. No statistics should be calculated if $n=2$.

5) a complete author checklist, which you can download from our author guidelines <https://www.embopress.org/page/journal/14693178/authorguide>. Please insert information in the checklist that is also

reflected in the manuscript. The completed author checklist will also be part of the RPF.

6) Please note that all corresponding authors are required to supply an ORCID ID for their name upon submission of a revised manuscript (<<https://orcid.org/>>). Please find instructions on how to link your ORCID ID to your account in our manuscript tracking system in our Author guidelines <<https://www.embopress.org/page/journal/14693178/authorguide#authorshipguidelines>>

12) All Materials and Methods need to be described in the main text using our 'Structured Methods' format, which is required for all research articles. According to this format, the Methods section includes a Reagents and Tools Table (listing key reagents, experimental models, software and relevant equipment and including their sources and relevant identifiers) followed by a Methods and Protocols section describing the methods using a step-by-step protocol format. The aim is to facilitate adoption of the methodologies across labs. More information on how to adhere to this format as well as a downloadable template (.docx) for the Reagents and Tools Table can be found in our author guidelines:

An example of a Method paper with Structured Methods can be found here: <https://www.embopress.org/doi/full/10.1038/s44320-024-00037-6#sec-4>

I look forward to seeing a revised form of your manuscript when it is ready.

Yours sincerely,

Yehu Moran
Editor
EMBO Reports

IMPORTANT Specific comments by our editorial assistant (checks format and adherence to author instructions. Please correct before submitting your revision):

CHARACTER COUNT: 31,006, 5 figures, Results & Discussion NOT combined

MANUSCRIPT FORMAT: NOT OK - figures included; I've removed the figures for the purpose of the revision checks but at revision, please be reminded not to include the figures in the manuscript file.

COI (conflict of interest)/DCIS: in, BUT it needs to be renamed to "Disclosure Statement and Competing Interests"

AC/CRedit: need to be removed from the ms and provided only via the journal system.

REFERENCE FORMAT: NOT OK - references need to be alphabetical, et al needs to be used after 10 author names; there is one DOI - DOIs should only be used for preprints and datasets that have not been published yet.

FUNDING INFO: the following grant numbers are missing in journal submission system and need to be inserted for the the National Natural Science Foundation of China: 32100335, 32070528

FIGURE CALLOUTS: 10 EV figures called out in the ms, BUT the figures are not uploaded as EV, they are provided in the Appendix file; if the figures will remain in the Appendix then all the callouts in the ms need to be updated accordingly; missing a callout for Table 1

DATASET EV LEGENDS: there are 5 EV tables, BUT they need to be uploaded as separate files; if the authors want to keep them in one file, then they should move them to the Appendix and change the nomenclature and callouts to Appendix Table S1. etc.

APPENDIX 1 FILE WITH ToC: included, BUT needs to be in PDF and needs to have page numbers; co-author Paul A. Garber is missing from the author list (btw, no need to keep the author list in the Appendix); as noted above, the callouts for the Appendix figures in the ms are not OK; if these figures need to be EV figures, we can accommodate up to 5 figures as EV figures and they need to be uplidd separately and the callouts in the ms need to be corrected accordingly

SYNOPSIS IMAGE: should be included.

SOURCE DATA: Authors need to provide links to the external repositories.

NOTES:

- The manuscript sections should be in the following order: Title page - Abstract & Keywords - Introduction - Results -Discussion - Methods - Data Availability - Acknowledgments - Disclosure Statement & Competing Interests - References -Figure Legends - Tables with legends - Expanded View Figure Legends.

Re-use of graph between Figure 4 and Appendix Figure S8. Is this re-use needed or necessary? Please check and explain.

Figure Legends - Comments

- The error bars are not defined in the legends of figures 2b-c. Please correct.

Referee #1:

I appreciate that the authors have made some changes to the manuscript and that they have added some new simulations, especially bottleneck simulations. However, my main concern remains. In particular, the manuscript attempts to draw connections between the proposed statistics and conservation status that are not warranted based on the analyses provided.

The analyses provided for the new bottleneck simulations suggest that LDcor will be affected by bottlenecks, and that the effects are time-dependent. LDcor is still increasing 10 generations after a bottleneck. Moreover, the results also depend on selection coefficients and the magnitude of the bottleneck. While I appreciate that several cases were analyzed, they all correspond to very small numbers of generations following a bottleneck, and it is not clear how to relate these temporally non-equilibrium results to any particular species.

One potential solution would be to fit a demographic model and selection model for each species and calculate LDcor to ask whether LDcor patterns can be interpreted in terms of the model parameters. The authors do simulate some models corresponding to humans, golden snub-nosed monkeys, and pandas. But the simulated model patterns do not correspond well with the empirical patterns. There are many reasons this could occur, some of which are discussed to some degree in the manuscript (e.g., epistasis acting on selected variants). Given the numerous factors that could affect these LD statistics, it is ultimately very unclear to me what may be driving any of the empirical patterns the authors observe. Suggesting that anything related to conservation status is relevant to the patterns seems like an extreme leap.

A key set of analyses compare LoF variants to missense or synonymous variants. The authors have shaded the regions of their "LD statistics of some mammal populations" in which LoF mutations are below synonymous variants. But there are very few LoF mutations in many of these datasets (Table 2) and no real statistics done to ask whether the differences are "real" or not.

In the abstract, it is stated that "Our results suggest that selection is not commonly relaxed in threatened mammals, which may explain why they persist in the wild despite very low population size." I'm not convinced that this can be argued from the analyses presented in the manuscript. There are no attempts in the manuscript to link specific demographic events to the empirical patterns observed. The few demographic models that were simulated do not yield patterns consistent with the empirical data. There is not any general way to link the statistics directly to the strength of negative selection, since they are clearly affected by both demographic processes and selection. There are only a handful of species investigated here and the patterns are not consistent enough between threatened and non-threatened species to say much of anything about the correspondence between the statistics and population vulnerability.

This issue is not only present in the abstract, but remains throughout other parts of the manuscript. The discussion has some speculation about the effects of ILS on the patterns, effects of habitat fragmentation, how to interpret differences in patterns between LoF variants and missense/synonymous variants, none of which are assessed by the manuscript in any direct fashion. Again, I appreciate the authors' attention to an important question. However, in my view the analyses presented in the revised manuscript are still not straightforward to interpret and the manuscript lacks a clear narrative.

Referee #2:

All comments and concerns have been appropriately addressed. I see no further dramatic changes to be done in this manuscript, which in my opinion is adequate for publication.

Referee #3:

The authors addressed the reviewers' comments by elaborating on the rationale for selecting specific mammal species and adding a comparison with gorillas to provide evolutionary insights.

They acknowledged the potential bias in assuming synonymous mutations are neutral and discussed the ongoing debate on this topic.

Detailed information on ancestral genome reconstruction was added, and the term "narrowing purified selection" was clarified to "relaxed negative selection."

The new statistical approaches, LDcor and LDcorabs, were introduced in the abstract, and consistent terminology ("negative selection") was used throughout the manuscript.

Classical methods (r2, D', NetLD) were introduced before comparison, and the R script was made available on GitHub.

However, the provided GitHub link is not described at all "<https://github.com/Chunyan-Hu/LDcor/tree/main>". They should include a README file in their GitHub repository to guide readers on reproducing their model "What is this test.txt file for example which is an empty file?" and specify which version of R is required to run their script.

Currently, they only mention in their manuscript that they use R without specifying the version, such as on line 124 of the manuscript "the calculation of NetLD23 was done using Rscript."

Lastly, the manuscript was polished for grammatical accuracy.

Detailed responses to editors and reviewers

All comments provided by Editor and reviewers are shown in gray italics, and our responses are shown in blue. All revisions in the manuscript are marked in red.

EDITOR COMMENTS

IMPORTANT Specific comments by our editorial assistant (checks format and adherence to author instructions. Please correct before submitting your revision):

CHARACTER COUNT: 31,006, 5 figures, Results & Discussion NOT combined

MANUSCRIPT FORMAT: NOT OK - figures included; I've removed the figures for the purpose of the revision checks but at revision, please be reminded not to include the figures in the manuscript file.

COI (conflict of interest)/DCIS: in, BUT it needs to be renamed to "Disclosure Statement and Competing Interests"

AC/CRedit: need to be removed from the ms and provided only via the journal system.

REFERENCE FORMAT: NOT OK - references need to be alphabetical, et al needs to be used after 10 author names; there is one DOI - DOIs should only be used for preprints and datasets that have not been published yet.

FUNDING INFO: the following grant numbers are missing in journal submission system and need to be inserted for the the National Natural Science Foundation of China: 32100335, 32070528

FIGURE CALLOUTS: 10 EV figures called out in the ms, BUT the figures are not uploaded as EV, they are provided in the Appendix file; if the figures will remain in the Appendix then all the callouts in the ms need to be updated accordingly; missing a callout for Table 1

DATASET EV LEGENDS: there are 5 EV tables, BUT they need to be uploaded as separate files; if the authors want to keep them in one file, then they should move them to the Appendix and change the nomenclature and callouts to Appendix Table S1. etc.

APPENDIX 1 FILE WITH ToC: included, BUT needs to be in PDF and needs to have page numbers; co-author Paul A. Garber is missing from the author list (btw, no need to keep the author list in the Appendix); as noted above, the callouts for the Appendix figures in the ms are not OK; if these figures need to be EV figures, we can accommodate up to 5 figures as EV figures and they need to be upload separately and the callouts in the ms need to be corrected accordingly

SYNOPSIS IMAGE: should be included.

SOURCE DATA: Authors need to provide links to the external repositories.

NOTES:

- The manuscript sections should be in the following order: Title page - Abstract & Keywords - Introduction - Results - Discussion - Methods - Data Availability - Acknowledgments - Disclosure Statement & Competing Interests - References - Figure Legends - Tables with legends - Expanded View Figure Legends.

Re-use of graph between Figure 4 and Appendix Figure S8. Is this re-use needed or necessary? Please check and explain.

Figure Legends - Comments

- The error bars are not defined in the legends of figures 2b-c. Please correct.

Response:

Thank you for your feedback. We have formatted the manuscript according to the journal's requirements:

1. All figures have been removed from the main text.
2. The "COI" section is now labeled "Disclosure Statement and Competing Interests."
3. AC/CRedit information has been removed from the manuscript and provided through the journal system.
4. We have updated the references to match the required format.
5. All funding information has been updated accordingly.
6. Figure callouts have been uploaded in the manuscript from "EV" to "Appendix." Tables 1-3 are included in the new submission, and the nomenclature for EV tables has been changed to "Appendix Table." The author list has been removed from the appendix file, which is now uploaded as a PDF.
7. We have included a synopsis image as requested.
8. All links to source data have been updated in the Appendix.
9. The manuscript sections have been reordered as per your reminder.
10. Appendix Figure S8 has been made distinct from Figure 4. In Figure 4, all filtered variants were retained regardless of their allele frequency spectrum (AFS). In Appendix Figure S8, variants were sampled from synonymous variants and missense variants with the same AFS as the LoF variants set (red line). This was done to demonstrate that our observations in Figure 4 held true even when AFS was controlled.
11. The definition of error bars has been added to the legends of Figures 2b and 2c.

We hope these changes meet your requirements. Please let us know if there are any further adjustments needed.

REFEREES COMMENTS

Referee #1

Comment 1-1:

I appreciate that the authors have made some changes to the manuscript and that they have added some new simulations, especially bottleneck simulations. However, my main concern remains. In particular, the manuscript attempts to draw connections between the proposed statistics and conservation status that are not warranted based on the analyses provided.

Response:

We sincerely appreciate your feedback on our manuscript. We would like to take this opportunity to explain our methodology regarding the developed LD statistics, which we believe offer several advantages in the evaluation of conservation status and conservation biology.

Advantages of Our Proposed Statistics

Reduction of Bias from Total Mutations:

The proposed statistics are not biased by the total number of mutations. Traditional measures of conservation status, such as genetic diversity, can sometimes be misleading (Cho et al, 2013; Zhang et al, 2015; Liu et al, 2018; Zhao et al, 2013; Zhou et al, 2016). For example, genetic diversity in endangered species like chimpanzees (*Pan troglodytes*), Tasmanian devils (*Sarcophilus harrisii*), tigers (*Panthera tigris*), lions (*Panthera leo*), and others can be similar to, or even higher than, that of non-threatened species. This happens because genetic diversity encompasses all mutations, including neutral ones, whose differences can obscure true conservation priorities. Our statistics, by normalizing deleterious mutations through LD of missense mutations, avoid this bias.

Consistency with Selection Coefficient:

Our proposed statistics correlate with selection coefficients, reflecting the evolutionary pressures on species. Metrics like LD_{cor} decreases with stronger negative selection, higher recombination rates, reduced synergistic epistasis, or more severe bottlenecks. Lower LD_{cor} and negative nLD_{cor} usually indicate greater negative linkage disequilibrium and lower conservation priority. Empirical data support this: for instance, the giant panda (currently VULNERABLE) shows negative nLD_{cor}s for LoF variants with mdac < 4, suggesting low but increasing evolutionary potential. Conversely, the tiger (currently ENDANGERED) displays positive nLD_{cor}s for LoF variants with mdac in the range, indicating decreasing evolutionary potential.

Feasibility for Cross-Species Comparison:

Our statistics can be used for cross-species comparisons, overcoming limitations faced by other methods that require homologous and similarly consequential mutations (Dusseux et al, 2021; Xue et al, 2015; Xie et al, 2022). While traditional assessments often presume variants are independent, our approach considers the global LD pattern of neutral, missense, and LoF variants, allowing comprehensive and comparative analyses across species (Muller, 1950; Charlesworth, 1990; B, 1998; Kondrashov, 1998; West et al, 1998; Bertorelle et al, 2022).

Based on the reasons mentioned above, we argue that our proposed LD statistics have significant potential for conservation biology. They can be used to measure the conservation priorities for all endangered animals effectively. To address your concerns, our co-author, Dr. Paul A. Garber, a conservation biologist, has reviewed and edited the introduction and discussion sections of the manuscript to ensure clarity and comprehensiveness.

Thank you once again for your feedback. Please review our revisions and let us know if they meet your satisfaction.

Comment 1-2:

The analyses provided for the new bottleneck simulations suggest that LDcor will be affected by bottlenecks, and that the effects are time-dependent. LDcor is still increasing 10 generations after a bottleneck. Moreover, the results also depend on selection coefficients and the magnitude of the bottleneck. While I appreciate that several cases were analyzed, they all correspond to very small numbers of generations following a bottleneck, and it is not clear how to relate these temporally non-equilibrium results to any particular species.

One potential solution would be to fit a demographic model and selection model for each species and calculate LDcor to ask whether LDcor patterns can be interpreted in terms of the model parameters. The authors do simulate some models corresponding to humans, golden snub-nosed monkeys, and pandas. But the simulated model patterns do not correspond well with the empirical patterns. There are many reasons this could occur, some of which are discussed to some degree in the manuscript (e.g., epistasis acting on selected variants). Given the numerous factors that could affect these LD statistics, it is ultimately very unclear to me what may be driving any of the empirical patterns the authors observe. Suggesting that anything related to conservation status is relevant to the patterns seems like an extreme leap.

Response:

Thank you for your feedback and further suggestions. Extremely long-duration bottleneck simulations have not been performed principally for the following reasons. First, prolonged generations and extended bottleneck events introduce significant stochasticity. Bottlenecks amplify the effects of genetic drift and distort the site frequency spectrum (SFS) of low-frequency alleles based on their intensity and duration (Gattepaille et al, 2013; Lucena - Perez et al, 2021; Nei et al, 1975; Garza & Williamson, 2001). The severity of a bottleneck can be expressed as $k = N_b/d$ (where N_b is the bottleneck population size, and d is the duration of the bottleneck) (Tang et al, 2010; Wright et al, 2005). Temporal spans up to 10 generations show different details than spans up to 100 times in severity (100~10000). An increased and prolonged bottleneck increases the stochasticity of linkage disequilibrium (LD) values, as suggested by higher standard error of the mean (s.e.m.) in simulations with more severe or longer-lasting bottlenecks (Fig 2B).

Second, when setting longer durations for bottleneck simulations, we presumed the simulated populations to be evolutionarily sustainable. This assumption faces challenges as accumulated risks are evident when these populations lose significant variability during long-duration bottlenecks in our simplified simulations, which may not reflect real-world scenarios.

Third, our LD correlation (LDcor) primarily measures the current status of LD concerning loss-of-function (LoF) by normalizing neutral and missense mutations. The simulations in this study were intended primarily to illustrate that more severe bottlenecks result in higher LDcor, rather than exploring all potential bottleneck scenarios.

Nonetheless, we added simulations with longer durations (i.e., 1, 10, and 100 generations) to assess a wider range of possibilities. All simulations spanned a length of 1 Mb with a mutation rate of 1.5×10^{-8} per generation per base pair. The recombination rate was kept constant at 1×10^{-8} per generation per base pair. Alleles were assumed to be additive ($h = 0.5$). To test how LD changes with increasing negative selection pressure, the selection strength on deleterious alleles

was set to 0, -10^{-4} , -10^{-3} , and -10^{-2} and -10^{-1} . Each parameter set was estimated with at least 240 replicates. The observed LD patterns (i.e., increased LD values with appropriate bottleneck duration) persisted when the bottleneck duration extended to 100 generations (Fig R1). This pattern was consistent even when the sequence length was increased from 1 Mb to 10 Mb or when the mutation rate was scaled from 1.5×10^{-8} to 1×10^{-7} . These new analyses suggest that LDcor is influenced by bottleneck duration within certain time limits, and these limits are affected by various factors.

Fig. R1: LDcor value under varying duration times of bottleneck (based on model 1). We simulated different scaled population size (from 10000 to 1000/700/400) as well as different duration time (0, 1, 10, and 100 generations). The dark gray line is the results of no-resampling at time 0 (i.e., time 00).

Apologies for any confusion caused in our demographic analyses. As we stated in the previous response, our LD statistics were intended to inform the assessment of genetic burden and conservation priority rather than demonstrate consistency with current conservation status as recommended by the IUCN (2022). Nevertheless, we observed a close alignment between our genetic statistics and the actual conservation expectations in the empirical analysis.

We appreciate your suggestion to use modeling for species to explore the drivers of empirical patterns. Numerous factors can influence LD statistics, and there is no one-to-one correspondence between demography and LD patterns (Figs. 1-2). For instance, both lower recombination rates and more severe bottlenecks tend to increase LDcor values. We did not aim to infer the exact demographic history from our statistics in either of the empirical or simulation analyses. Instead, we selected species with well-studied demographic histories to validate the utility of our statistics in indicating evolutionary potential. The simulations relied on well-documented and simplified demographic models, which are not universally applicable. Therefore, we conducted simulations as examples for the well-studied golden snub-nosed monkey and giant panda to illustrate the potential distribution of LDcor in under estimating demography (Fig. 5).

We may disagree with your judgment of "inconsistency". Although the exact LDcor values and nLDcor values from species' simulations differed from those of the empirical values, we found evidence of empirical patterns to a certain extent from the distribution of LDcor under appropriate simulations. For example, in the golden snub-nosed monkey simulation, LDcor distributions for variants with low selection coefficients ($s=-0.0001$) were similar to those under neutrality ($s=0$). This is somewhat consistent with the observed pattern of higher LDcor in LoF variants in the empirical data. Another example is the giant panda simulation, where LDcor distributions varied distinctly among different selection coefficients. In summary, despite the LD values not being identical, our simulations still captured the general outline of empirical patterns.

The differences in LDcor values likely arise for three reasons. First, accurately determining the exact demography and evolutionary parameters is challenging. We calculate LDcor for variants with the same selection coefficient during simulations, whereas LoF variants in empirical data have varying, inconsistent, and often unknown selection coefficients. Second, the count of variants, SNP frequency spectrum, and LD distribution can still vary even with known evolutionary history and parameters. We observed overlap between the distribution of LDcor under adjacent selection coefficients in simulations when all other parameters were constant. The observed pattern is just one of many possible patterns under a specified demography (we only presented the mean and standard error of LDcor in the manuscript). Third, it is theoretically possible to resample variants from simulation results to match the allele frequency spectrum of empirical data, but this requires extensive simulations, which are costly. Therefore, the differences in exact LDcor values are reasonable.

Comment 1-3:

A key set of analyses compare LoF variants to missense or synonymous variants. The authors have shaded the regions of their "LD statistics of some mammal populations" in which LoF mutations are below synonymous variants. But there are very few LoF mutations in many of these datasets (Table 2) and no real statistics done to ask whether the differences are "real" or not.

Response:

Thank you for pointing this out. Indeed, very few loss-of-function (LoF) mutations were observed in our data, which is due to the inclusion of only mutations with derived allele counts in the range of [2,5] (MDAC=5). For instance, fewer than 100 LoF mutations (i.e., the "Nloci" column) were found in more than half of the tested populations. The very low frequencies of rare LoF mutations (i.e., in the range of [1,5]) have been previously presented by Sohail et al, 2017.

We paid close attention to controlling for potential bias caused by allele count and allele frequency in different mutation sets (e.g., LoF/missense/synonymous). To address this, we:

1. Regulated the effect caused by these differences by defining our statistics based on the mean linkage disequilibrium (LD) of all rare SNP pairs (e.g., MDAC \leq 5) and normalizing the LD value of each SNP pair by the standard deviations of both SNPs. This approach is one of the advantages discussed in the methods section (lines 203-217, 286-292, 448-460).

2. Considering the effect of the SNP frequency spectrum (SFS) in both the supplementary materials and the discussion, we hypothesized that the difference in SFS may be indicative of

evolutionary processes. Our observed patterns remained consistent even after controlling for SFS (Appendix Figures S2, S8, lines 120-124, 220-226, 293-301,607-613).

In summary, the count differences between LoF mutations and synonymous mutations were normalized, suggesting that the detected differences in LD values are robust and reliable.

Appendix Figure S8: LDcor and LDcorabs values of empirical data when allele frequency spectrum is controlled.

Appendix Figure S2: Mean LDcor and LDcorabs values with different selection coefficient when allele frequency spectrum is controlled.

(A) We resampled variants from the 0 simulations with the same allele frequency spectrum of -0.01 simulations and compared LDcor. The blue solid line represents LD of 0 simulations (named as S), the purple dotted line represents -0.01 simulations (named as D), and the red dotted line with represents 0 simulations but with the same allele frequency spectrum of -0.01 simulations (named as SR). The LDcor of SR is always between that of S and D.

(B) The LDcorabs values in S, D, and SR under varying max derived allele count.

Comment 1-4:

In the abstract, it is stated that "Our results suggest that selection is not commonly relaxed in threatened mammals, which may explain why they persist in the wild despite very low population size." I'm not convinced that this can be argued from the analyses presented in the manuscript. There are no attempts in the manuscript to link specific demographic events to the empirical patterns observed. The few demographic models that were simulated do not yield patterns consistent with the empirical data. There is not any general way to link the statistics directly to the strength of negative selection, since they are clearly affected by both demographic processes and selection. There are only a handful of species investigated here and the patterns are not consistent enough between threatened and non-threatened species to say much of anything about the correspondence between the statistics and population vulnerability.

This issue is not only present in the abstract, but remains throughout other parts of the manuscript. The discussion has some speculation about the effects of ILS on the patterns, effects of habitat fragmentation, how to interpret differences in patterns between LoF variants and missense/synonymous variants, none of which are assessed by the manuscript in any direct fashion.

Response:

We apologize for overstating the results.. We have revised the text accordingly. We agree with your idea that there is no general method to directly link the statistics to the strength of

negative selection, as they are influenced by both demographic processes and selection. However, it is observed that intensified negative selection generally induces negative linkage disequilibrium (LD), setting it apart from other influencing factors to some degree. Hence, we can still infer strong negative selection from negative LDcor and negative nLDcor, despite the inability to precisely the effects of selection or other factors on the value of nLDcor (i.e., the difference in LDcor values between loss-of-function (LoF) and synonymous mutations).

Among the factors we have tested, limited recombination rate, small population size, and the presence of population bottlenecks tend to increase LDcor by resulting in positive LD. Conversely, intensified negative selection and synergistic epistasis tend to produce negative LD and LDcor. For a given selection coefficient or set of variants, a higher LDcor value typically indicates one or more of the following: a lower recombination rate, lower levels of synergistic epistasis, a more severe bottleneck, longer bottleneck duration, and/or shigher conservation priority. Similarly, for pairs with the same selection coefficient in nLDcor (i.e., LDcor(s1) - LDcor(s2)), a negative nLDcor value generally suggests one or more of these same conditions and higher conservation priority. Moreover, by detecting and comparing global linkage disequilibrium of deleterious mutations using unphased genotypes, our LD statistics differs from those methods used for measuring genetic diversity and genetic burden, as well as from other LD statistics (Table S8, lines 271-301).

In this context, we assumed LDcor/nLDcor can indicate genetic burden or, to some degree, population vulnerability.

We determined the effects of ILS and habitat fragmentation due to their analogous roles to factors that that link directly to LD patterns. Using ILS as an example, we examined the stochastic transfer of segments during speciation. The stochastic, non-random sampling of ILS is similar to founder effect, where founder bias directly influences LD patterns and conservation priorities. When examining habitat fragmentation, we noted both reduced population sizes and reduced effective population sizes in isolated populations, akin to a bottleneck effect. We have streamlined these sections in the revision to enhance clarity and focus.

Table S3. The comparison of several LD statistics.

Index	D	D' / r²	NetLD	LDcor/LDcorabs	nLDcor
object	pairwise loci	pairwise loci	loci set	loci set	loci set
normalized	no	yes	no	yes	yes
directional	yes	no	yes	yes/no	yes
comparable between variant sets	no	yes	no	yes	yes
comparable between species	no	no	no	no	yes

Figure 1B: LDcor and nLDcor with different selection coefficients, recombination rates, and demographic models.

Appendix Figure S4: Simulation of nLDcor with different levels of synergistic epistasis, bottlenecks, demography, and max derived allele frequency.

Comment 1-5:

Again, I appreciate the authors' attention to an important question. However, in my view the analyses presented in the revised manuscript are still not straightforward to interpret and the manuscript lacks a clear narrative.

Response:

Thank you once again for your 1 comments. We hope our response and new analyses have alleviated your concerns. If you have any further specific suggestions, please let us know.

Referee #2

All comments and concerns have been appropriately addressed. I see no further dramatic changes to be done in this manuscript, which in my opinion is adequate for publication.

Response:

Thank you very much.

Referee #3

The authors addressed the reviewers' comments by elaborating on the rationale for selecting specific mammal species and adding a comparison with gorillas to provide evolutionary insights. They acknowledged the potential bias in assuming synonymous mutations are neutral and discussed the ongoing debate on this topic.

Detailed information on ancestral genome reconstruction was added, and the term "narrowing purified selection" was clarified to "relaxed negative selection."

The new statistical approaches, LDcor and LDcorabs, were introduced in the abstract, and consistent terminology ("negative selection") was used throughout the manuscript.

Classical methods (r^2 , D' , NetLD) were introduced before comparison, and the R script was made available on GitHub.

However, the provided GitHub link is not described at all

"<https://github.com/Chunyan-Hu/LDcor/tree/main>". They should include a README file in their GitHub repository to guide readers on reproducing their model "What is this test.txt file for example which is an empty file?" and specify which version of R is required to run their script.

Currently, they only mention in their manuscript that they use R without specifying the version, such as on line 124 of the manuscript "the calculation of NetLD23 was done using Rscript."

Lastly, the manuscript was polished for grammatical accuracy.

Response:

Thank you very much for your support and edits. We have now added a 'README.txt' file to guide the preparation and usage of our statistics, including the feasible environment configuration and command lines.

Reference

- Akçakaya HR, Butchart SHM, Mace GM, Stuart SN & Hilton-Taylor C (2006) Use and misuse of the IUCN Red List Criteria in projecting climate change impacts on biodiversity. *Glob Chang Biol* 12: 2037–2043
- B C (1998) The effect of synergistic epistasis on the inbreeding load. *Genet Res* 71: 85–89
- Bertorelle G, Raffini F, Bosse M, Bortoluzzi C, Iannucci A, Trucchi E, Morales HE & van Oosterhout C (2022) Genetic load: genomic estimates and applications in non-model animals. *Nat Rev Genet* 23: 492–503
- Brüniche-Olsen A, Kellner KF & DeWoody JA (2019) Island area, body size and demographic history shape genomic diversity in Darwin’s finches and related tanagers. *Mol Ecol* 28: 4914–4925
- Cardoso P, Borges PAV, Triantis KA, Ferrández MA & Martín JL (2011) Adapting the IUCN Red List criteria for invertebrates. *Biol Conserv* 144: 2432–2440
- Charlesworth B (1990) Mutation-selection balance and the evolutionary advantage of sex and recombination. *Genet Res* 55: 199–221
- Cho YS, Hu L, Hou H, Lee H, Xu J, Kwon S, Oh S, Kim H-M, Jho S, Kim S, *et al* (2013) The tiger genome and comparative analysis with lion and snow leopard genomes. *Nat Commun* 4: 2433
- Coates D (2018) Strategic Plan for Biodiversity (2011–2020) and the Aichi Biodiversity Targets. In *The Wetland Book* pp 493–499. Dordrecht: Springer Netherlands
- Dussex N, Morales HE, Grossen C, Dalén L & van Oosterhout C (2023) Purging and accumulation of genetic load in conservation. *Trends Ecol Evol* 38: 961–969
- Dussex N, van der Valk T, Morales HE, Wheat CW, Díez-del-Molino D, von Seth J, Foster Y, Kutschera VE, Guschanski K, Rhie A, *et al* (2021) Population genomics of the critically endangered kākāpō. *Cell Genomics* 1: 100002
- Garza JC & Williamson EG (2001) Detection of reduction in population size using data from microsatellite loci. *Mol Ecol* 10: 305–318
- Gattepaille LM, Jakobsson M & Blum MG (2013) Inferring population size changes with sequence and SNP data: lessons from human bottlenecks. *Heredity (Edinb)* 110: 409–419
- Hansson B & Westerberg L (2002) On the correlation between heterozygosity and fitness in natural populations. *Mol Ecol* 11: 2467–2474
- Hayward MW, Child MF, Kerley GIH, Lindsey PA, Somers MJ & Burns B (2015) Ambiguity in guideline definitions introduces assessor bias and influences consistency in IUCN Red List status assessments. *Front Ecol Evol* 3
- IUCN. 2022. The IUCN Red List of Threatened Species. Version 2022-2. <https://www.iucnredlist.org>. Accessed on 09 December 2022.
- Kondrashov AS (1998) Measuring spontaneous deleterious mutation process. *Genetica* 102–103: 183–97
- Liu Y-C, Sun X, Driscoll C, Miquelle DG, Xu X, Martelli P, Uphyrkina O, Smith JLD, O’Brien SJ & Luo S-J (2018) Genome-Wide Evolutionary Analysis of Natural History and Adaptation in the World’s Tigers. *Curr Biol* 28: 3840–3849.e6
- Lucena-Perez M, Kleinman-Ruiz D, Marmesat E, Saveljev AP, Schmidt K & Godoy JA (2021) Bottleneck-associated changes in the genomic landscape of genetic diversity in wild lynx populations. *Evol Appl* 14: 2664–2679
- Lynch M, Conery J & Burger R (1995) Mutation Accumulation and the Extinction of Small

- Populations. *Am Nat* 146: 489–518
- Muller HJ (1950) Our load of mutations. *Am J Hum Genet* 2: 111–76
- Nei M, Maruyama T & Chakraborty R (1975) THE BOTTLENECK EFFECT AND GENETIC VARIABILITY IN POPULATIONS. *Evolution (N Y)* 29: 1–10
- Nonić M & Šijačić-Nikolić M (2021) Genetic Diversity: Sources, Threats, and Conservation. In pp 421–435.
- Van Oosterhout C (2020) Mutation load is the spectre of species conservation. *Nat Ecol Evol* 4: 1004–1006
- Sohail M, Vakhrusheva OA, Sul JH, Pulit SL, Francioli LC, Van Den Berg LH, Veldink JH, De Bakker PIW, Bazykin GA, Kondrashov AS, *et al* (2017) Negative selection in humans and fruit flies involves synergistic epistasis. *Science (80-)* 356: 539–542
- Tang H, Sezen U & Paterson AH (2010) Domestication and plant genomes. *Curr Opin Plant Biol* 13: 160–166
- Trull N, Böhm M & Carr J (2018) Patterns and biases of climate change threats in the IUCN Red List. *Conserv Biol* 32: 135–147
- West SA, Peters AD & Barton NH (1998) Testing for epistasis between deleterious mutations. *Genetics* 149: 435–44
- Wright SI, Bi IV, Schroeder SG, Yamasaki M, Doebley JF, McMullen MD & Gaut BS (2005) The Effects of Artificial Selection on the Maize Genome. *Science (80-)* 308: 1310–1314
- Xie H-X, Liang X-X, Chen Z-Q, Li W-M, Mi C-R, Li M, Wu Z-J, Zhou X-M & Du W-G (2022) Ancient Demographics Determine the Effectiveness of Genetic Purging in Endangered Lizards. *Mol Biol Evol* 39
- Xue Y, Prado-Martinez J, Sudmant PH, Narasimhan V, Ayub Q, Szpak M, Frandsen P, Chen Y, Yngvadottir B, Cooper DN, *et al* (2015) Mountain gorilla genomes reveal the impact of long-term population decline and inbreeding. *Science (80-)* 348: 242–245
- Zhang W, Luo Z, Zhao M & Wu H (2015) High genetic diversity in the endangered and narrowly distributed amphibian species *Leptobranchium leishanense*. *Integr Zool* 10: 465–81
- Zhao S, Zheng P, Dong S, Zhan X, Wu Q, Guo X, Hu Y, He W, Zhang S, Fan W, *et al* (2013) Whole-genome sequencing of giant pandas provides insights into demographic history and local adaptation. *Nat Genet* 45: 67–71
- Zhou X, Meng X, Liu Z, Chang J, Wang B, Li M, Wengel PO, Tian S, Wen C, Wang Z, *et al* (2016) Population Genomics Reveals Low Genetic Diversity and Adaptation to Hypoxia in Snub-Nosed Monkeys. *Mol Biol Evol* 33: 2670–2681

Manuscript number: EMBOR-2023-58241V3

Title: Genetic linkage disequilibrium of deleterious mutations in threatened mammals

Author(s): Chunyan Hu, Gaoming Liu, Zhan Zhang, Qi Pan, Xiaoxiao Zhang, Weiqiang Liu, Zihao Li, Meng Li, Pingfen Zhu, Ting Ji, Paul Garber, and Xuming Zhou

Dear Dr. Zhou

Thank you for your patience while we have reviewed your revised manuscript. As you will see from the reports below, the referees are now all positive about its publication in EMBO Reports. I am therefore writing with an 'accept in principle' decision, which means that I will be happy to accept your manuscript for publication once a few minor issues/corrections have been addressed, as follows:

Referee #1 is still concerned regarding some of the phrasing and sentences that you use that are overstating some findings or are just somewhat ambiguous. I tend to agree on this. Please make these corrections so we can proceed to official acceptance of the manuscript.

Once you have made these minor revisions, please use the following link to submit your corrected manuscript:

Link Not Available

If all remaining corrections have been attended to, you will then receive an official decision letter from the journal accepting your manuscript for publication in the next available issue of EMBO Reports. This letter will also include details of the further steps you need to take for the prompt inclusion of your manuscript in our next available issue.

Yours sincerely,

Yehu Moran
Academic Editor
EMBO Reports

Referee #1:

I appreciate the authors inclusion of some new simulations and their attention to prior critiques. I have a few remaining comments about the language in the manuscript. Specifically, the claim that "these newly developed statistics have potential applications for conservation biology" (made in the abstract) is not justified by the data and analyses presented. There are additional phrases and claims in the manuscript that I highlight below, which I suggest removing (see below).

Main comments:

In their reply, the authors write "The differences in LDcor values likely arise for three reasons. First, accurately determining the exact demography and evolutionary parameters is challenging. We calculate LDcor for variants with the same selection coefficient during simulations, whereas LoF variants in empirical data have varying, inconsistent, and often unknown selection coefficients. Second, the count of variants, SNP frequency spectrum, and LD distribution can still vary even with known evolutionary history and parameters. We observed overlap between the distribution of LDcor under adjacent selection coefficients in simulations when all other parameters were constant. The observed pattern is just one of many possible patterns under a specified demography (we only presented the mean and standard error of LDcor in the manuscript). Third, it is theoretically possible to resample variants from simulation results to match the allele frequency spectrum of empirical data, but this requires extensive simulations, which are costly". I broadly agree with most of these reasons that results could differ between the simulations and observed data. These are also the exact reasons that interpreting empirically observed LDcor in terms of conservation status will not be straightforward. All of these factors and more will affect LDcor in a variety of ways. I think this could be clarified substantially in the manuscript. The section on confounders could be more direct about these interpretation challenges.

I appreciate the resampling analysis performed in Fig S8, and apologize if I missed this on a previous review. However, the quartile ranges around the median are not visible on the plot, which makes it less useful in interpreting the effect of sampling variance. Is this because the windows are not actually plotted, or is it because they are very small and not visible? The latter seems unlikely, but should be explicitly stated if so. In general, it would be more conservative to plot a broader window (e.g., 95% of the probability mass) rather than just the middle 50%.

I also suggest the following specific changes to remove ambiguous or overstated claims:

Abstract: I suggest removing the line "these newly developed statistics have potential applications for conservation biology". The abstract needs to avoid speculative statements, and the current manuscript does not support this statement.

Line 111: the phrase "and can be used to set conservation priorities and to design conservation strategies to protect a taxonomically diverse set of threatened mammals" is overstated. Consider rephrasing to "and might have applications in conservation biology when used in concert with other sources of data" or something similar. As it is currently phrased, there is no support for this definitive statement based on the manuscript.

Line 189: the phrase "The elimination of close loci (loci with distances of no more than 100 bp, or 1000 bp) resulted in smaller LD_{cor} values, implying Hill-Robertson effects may not be the primary source of negative LD." needs more explanation. Same on line 331.

Line 253: "This highlights the varied impact of negative selection across species and suggests nuanced evolutionary pressures at play." I think the MS should be more direct and simply state that this shows that LD_{cor} cannot be used in isolation to interpret conservation risk.

Line 261: "This suggests a potential over-accumulation of linkage among mildly selected variants", I'm not sure what the point is here? I would suggest using a different word than "over-accumulation" or a more extensive explanation.

Line 376: The phrase "This genetic evidence underpins the recent change in the conservation status of the giant panda from Endangered to Vulnerable" is unclear, I'm not sure what this is saying. If the claim is that LD_{cor} can be used to interpret the change in status then I don't think I agree, but perhaps that is not what is being said here?

All editorial and formatting issues were resolved by the authors.

Xuming Zhou
Chinese Academy of Sciences
B313 1-5 Beichen West Road
Beijing, Beijing 100101
China

Dear Dr. Zhou,

I am very pleased to accept your manuscript for publication in the next available issue of EMBO Reports. Thank you for your contribution to our journal.

Yours sincerely,

Yehu Moran
Editor
EMBO Reports
